# A spectral method for assessing and combining multiple data visualizations

Rong Ma [1], Eric D. Sun [2] & James Zou [2]

Dimension reduction is an indispensable part of modern data science, and many algorithms have been developed. However, different algorithms have their own strengths and weaknesses, making it important to evaluate their relative performance, and to leverage and combine their individual strengths. This paper proposes a spectral method for assessing and combining multiple visualizations of a given dataset produced by diverse algorithms. The proposed method provides a quantitative measure – the visualization eigenscore – of the relative performance of the visualizations for preserving the structure around each data point. It also generates a consensus visualization, having improved quality over individual visualizations in capturing the underlying structure. Our approach is flexible and works as a wrapper around any visualizations. We analyze multiple real-world datasets to demonstrate the effectiveness of the method. We also provide theoretical justifications based on a general statistical framework, yielding several fundamental principles along with practical guidance.

Data visualization and dimension reduction is a central topic in statistics and data science, as it facilitates intuitive understanding and global views of high-dimensional datasets and their underlying structural patterns through a low-dimensional embedding of the data[1,2]. The past decades have witnessed an explosion in machine learning algorithms for data visualization and dimension reduction. Many of them, such as Laplacian eigenmap[3], kernel principal component analysis (kPCA)[4], t-SNE[5], and UMAP[6], have been regarded as indispensable tools and state-of-art techniques for generating graphics in academic and professional writings[7], and for exploratory data analysis and pattern discovery in many research disciplines, such as astrophysics[8], computer vision[9], genetics[10], molecular biology[11], especially in single-cell transcriptomics[12], among others.

However, the wide availability and functional diversity of data visualization methods also brings forth new challenges to data analysts and practitioners[13,14]. On the one hand, it is critically important to determine among the extensive list which visualization method is most suitable and reliable for embedding a given dataset. In fact, even for a single visualization method, such as t-SNE or UMAP, oftentimes there are multiple tuning parameters to be determined by the users, and different tuning parameters may lead to distinct visualizations[15,16].

Thus, for a given dataset, selecting the most suitable visualization method and along with its tuning parameters calls for a method that provides quantitative and objective assessment of different visualizations of the dataset. On the other hand, as different methods are usually based on distinct ideas and heuristics, they would generate qualitatively diverse visualizations of a dataset, each containing important features about the data that are possibly unique to the visualization method. Meanwhile, due to the noisiness and high-dimensionality of many real-world datasets, their low-dimensional visualizations necessarily contain distortions from the underlying true structures, which again may vary from one visualization to another. It is therefore of substantial practical interest to combine strengths and reach a consensus among multiple data visualizations, in order to obtain an even better meta-visualization of the data that captures the most information and is least susceptible to the distortions. Naturally, a meta-visualization would also save practitioners from painstakingly selecting a single visualization method among many.

Quantitative assessment of dimension reduction and data visualization algorithms have been studied extensively. For example, many evaluation methods are based on distortion measures from metric geometry[17–20], whereas some other methods rely on information-

[1]Department of Statistics, Stanford University, Stanford, CA, USA. [2]Department of Biomedical Data Science, Stanford University, Stanford, CA, USA.
✉e-mail: jamesz@stanford.edu

theoretic precision-recall measures[21,22], co-ranking structure[23], or graph-based criteria[16,24]. See also recent reviews by Bertini et al.[25], Nonato and Aupetit[13] and Espadoto et al.[14]. However, most of these existing methods evaluate data visualizations by comparing them directly with the original dataset, without accounting for its noisiness. The thus obtained assessment may suffer from intrinsic bias due to ignorance of the underlying true structures, only approximately represented by the noisy observations.

Compared to the quantitative assessment of data visualizations, there is a scarcity of meta-visualization methods that combine strengths of multiple data visualizations. Pagliosa et al.[26] proposed an interactive method that assesses and combines different multidimensional projection methods via a convex combination technique. However, for supervised learning tasks such as classification, there is a long history of research on designing and developing meta-classifiers that combine multiple classifiers[27–31]. Compared with meta-classification, the main difficulty of meta-visualization lies in the identification of a common space to properly align multiple visualizations, or low-dimensional embeddings, whose scales and coordinate bases may drastically differ from one to another. Moreover, unlike many meta-classifiers, which combines presumably independent classifiers trained over different datasets, a meta-visualization procedure typically relies on multiple visualizations of the same dataset, and therefore has to deal with more complicated correlation structure among the visualizations. The current study provides the first meta-visualization method that can flexibly combine any number of visualizations, and has interpretable and provable performance guarantee.

Here we present a spectral method for assessing and combining multiple visualizations of a given dataset produced by diverse algorithms, allowing for different settings of tuning parameters for individual algorithms. The proposed method provides a quantitative measure – the visualization eigenscore – of the relative performance of the visualizations for preserving the structure around each data point.

It also generates a consensus visualization, having improved quality over individual visualizations in capturing the underlying structure. Our approach is flexible and works as a wrapper around any visualizations. In particular, our approach only needs access to the low-dimensional embeddings rather than the raw data; as a result, the users can use our method even if they don't have access to the original data, which is often the case.

## Results

### Overview of the method

Specifically, the proposed method takes as input a collection of visualizations, or low-dimensional embeddings of a dataset, hereafter referred as candidate visualizations, and summarizes each visualization by a normalized pairwise-distance matrix among the samples. With respect to each sample in the dataset, we construct a comparison matrix from these normalized distance matrices, characterizing the local concordance between each pair of candidate visualizations. Based on eigen-decomposition of the comparison matrices, we propose a quantitative measure, referred as visualization eigenscore, that quantifies the relative performance of the candidate visualizations in a sample-wise manner, reflecting their local concordance with the underlying low-dimensional structure contained in the data. To obtain a meta-visualization, the candidate visualizations are combined together into a meta-distance matrix, defined as a row-wise weighted average of those normalized distance matrices, using the corresponding eigenscores as the weights. The meta-distance matrix is then used to produce a meta-visualization, based on an existing method such as UMAP or kPCA, which is shown to be more reliable and more informative compared to individual candidate visualizations. Our method is schematically summarized in Fig. 1 and Algorithm 1, and detailed in Method section. The thus obtained meta-visualization reflects a joint perspective aggregating various aspects of the data that are oftentimes captured separately by individual candidate visualizations.

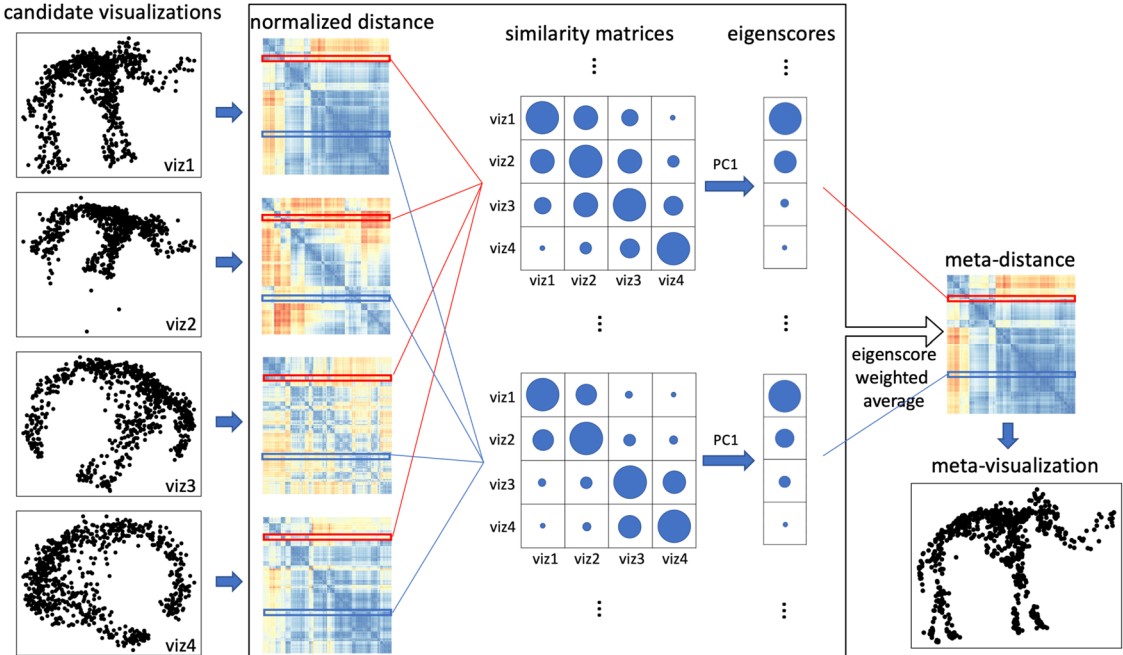

**Fig. 1 | A graphical illustration of the proposed method.** The algorithm takes as input the normalized pairwise distance matrices associated to a collection of candidate visualizations (viz1 to viz4) of a dataset. For each sample of the dataset, we compute the similarity matrix between the rows of the normalized distance matrices associated to the sample (rows highlighted in the same color), and then define the corresponding eigenscores as the first eigenvector of the similarity matrix. The size of the circles in the similarity matrices and the vectors of eigenscores indicate the magnitude of the entries (assumed to be non-negative). The meta-distance matrix is defined such that its rows are the eigenscore-weighted average of the rows in the normalized distance matrices. The meta-distance leads to a meta-visualization, expected to be more concordant with the underlying true structure than individual candidate visualizations.

Numerically, through extensive simulations and analysis of multiple real-world datasets with diverse underlying structures, we show the effectiveness of the proposed eigenscores in assessing and ranking a collection of candidate visualizations, and demonstrate the superiority of the final meta-visualization over all the candidate visualizations in terms of identification and characterization of these structural patterns. To achieve a deeper understanding of the proposed method, we also develop a formal statistical framework, that rigorously justifies the proposed scoring and meta-visualization method, providing theoretical insights on the fundamental principles behind the empirical success of the method, along with its proper interpretations, and guidance on practice.

The main features of the method can be summarized as follows:

- We propose a computationally efficient spectral method for assessing and combining multiple data visualizations. The method is generic and easy to implement: it does not require knowledge of the original dataset, and can be applied to a large number of data visualizations generated by diverse methods.
- For any collection of visualizations of a dataset, our method provides a quantitative measure – eigenscore – of the relative performance of the visualizations for preserving the structure around each data point. The eigenscores are useful on their own rights for assessing the local and global reliability of a visualization in representing the underlying structures of the data, and in guiding selection of hyper-parameters.
- The proposed method automatically combines strengths and ameliorates weakness (distortions) of the candidate visualizations, leading to a meta-visualization, which is provably better than all the candidate visualizations under a wide range of settings. We show that the meta-visualization is able to capture diverse intrinsic structures, such as clusters, trajectories, and mixed low-dimensional structures, contained in noisy and high-dimensional datasets.
- We establish rigorous theoretical justifications of the method under a general signal-plus-noise model in the large-sample limit. We prove the convergence of the eigenscores to certain underlying true concordance measures, the guaranteed performance of the meta-visualization and its advantages over alternative methods, its robustness against possible adversarial candidate visualizations, along with their conditions, interpretations, and practical implications.

## Simulation Studies: Visualizing Noisy Low-Dimensional Structures

To demonstrate the wide range of applicability and the empirical advantage of the proposed method, we consider visualization of three families of noisy datasets, each containing a distinct low-dimensional structure as its underlying true signal. We assess performance of the eigenscores and the quality of the resulting meta-distance matrix based on 16 candidate visualizations produced by multiple visualization methods.

For a given sample size $n$, we generate $p$-dimensional noisy observations $\{\mathbf{Y}_i\}_{1 \le i \le n}$ from the signal-plus-noise model $\mathbf{Y}_i = \mathbf{Y}_i^* + \mathbf{Z}_i$, where $\{\mathbf{Y}_i^*\}_{1 \le i \le n}$ are the underlying noiseless samples (signals), and $\{\mathbf{Z}_i\}_{1 \le i \le n}$ are the random noises. Specifically, we generate true signals $\{\mathbf{Y}_i^*\}_{1 \le i \le n}$ from various low-dimensional structures isometrically embedded in the $p$-dimensional Euclidean space. Each of the low-dimensional structures lie in some $r$-dimensional linear subspace, and is subject to an arbitrary rotation in $\mathbb{R}^p$, so that these signals are generally $p$-dimensional vectors with dense (nonzero) coordinates. Then we generate $i.i.d.$ noise vector $\mathbf{Z}_i$ from the standard multivariate normal distribution $\mathcal{N}(\mathbf{0}, \mathbf{I}_p)$, and use the $p$-dimensional noisy vector $\mathbf{Y}_i = \mathbf{Y}_i^* + \mathbf{Z}_i$ as the final observed data. In this way, we simulated noisy observations $\{\mathbf{Y}_i\}_{1 \le i \le n}$ of an intrinsically $r$-dimensional structure. For our simulations, for some given signal-to-noise ratio (SNR) parameter

$\theta > 0$, we generate $\{\mathbf{Y}_i^*\}_{1 \le i \le n}$ uniformly from each of the following three structures:

(i)   Finite point mixture with $r = 5$: $\{\mathbf{Y}_i^*\}_{1 \le i \le n}$ are independently sampled from the discrete set $\{\boldsymbol{\gamma}_1, \boldsymbol{\gamma}_2, \ldots, \boldsymbol{\gamma}_{r+1}\} \subset \mathbb{R}^p$ with equal probability, where $\boldsymbol{\gamma}_i$'s are arbitrary orthogonal vectors in $\mathbb{R}^p$ with the same length, i.e., $\|\boldsymbol{\gamma}_i\|_2 = \theta$ for $1 \le i \le r + 1$.

(ii)  Smiley face with $r = 2$: $\{\mathbf{Y}_i^*\}_{1 \le i \le n}$ are generated independently and uniformly from a two-dimensional smiley face structure (Supplementary Fig. 1 left) of diameter $\theta$, isometrically embedded in $\mathbb{R}^p$ and subject to an arbitrary rotation.

(iii) Mammoth manifold with $r = 3$: $\{\mathbf{Y}_i^*\}_{1 \le i \le n}$ are generated independently uniformly from a three-dimensional mammoth manifold (Supplementary Fig. 1 right) of diameter $\theta$, isometrically embedded in $\mathbb{R}^p$ and subject to an arbitrary rotation.

The thus generated datasets cover diverse structures including Gaussian mixture clusters (i), mixed-type nonlinear clusters (ii), and a connected smooth manifold (iii). As a result, the first family of datasets was set to have $p = 500$ and $n = 900$, and were obtained by fixing various values of the SNR parameter $\theta$, and generating $\mathbf{Y}_i^* \in \mathbb{R}^p$ from the above setting (i) to obtain the noisy dataset $\{\mathbf{Y}_i\}_{1 \le i \le n}$ as described above. Similarly, the second and the third families of datasets were obtained by drawing $\mathbf{Y}_i^* \in \mathbb{R}^p$ from the above settings (ii) and (iii), respectively, and generating datasets $\{\mathbf{Y}_i\}_{1 \le i \le n}$ with $p = 300$ and $n = 500$, for various values of $\theta$.

For each dataset $\{\mathbf{Y}_i\}_{1 \le i \le n}$, we consider 12 existing data visualization tools including principal component analysis (PCA), multidimensional scaling (MDS), Kruskal's non-metric MDS (iMDS)[32], Sammon's mapping (Sammon)[33], locally linear embedding (LLE)[34], Hessian LLE (HLLE)[35], isomap[36], kPCA, Laplacian eigenmap (LEIM), UMAP, t-SNE and PHATE[37]. For methods such as kPCA, t-SNE, UMAP and PHATE, that require tuning parameters, we consider two different settings (Supplementary file Section A.2) of tuning parameters for each method, denoted as kPCA1 and kPCA2, etc. Therefore, for each dataset we obtain $K = 16$ candidate visualizations corresponding to different combinations of visualization tools and tuning parameters. Applying our proposed method, we obtain eigenscores $\{\widehat{\mathbf{s}}_i\}_{1 \le i \le n}$ for the candidate visualizations. We also compare two meta-distances based on the 16 visualizations, which are, the proposed spectral meta-distance matrix (meta-spec) based on the eigenscores, and the naive meta-distance matrix (meta-aver) assigning equal weights to all the candidate visualizations, as in (4).

To evaluate the proposed eigenscores, for each setting and each $i \in \{1, 2, \ldots, n\}$, we compute $\cos \angle (\widehat{\mathbf{s}}_i, \mathbf{s}_i) := \frac{(\widehat{\mathbf{s}}_i)^\top \mathbf{s}_i}{\|\widehat{\mathbf{s}}_i\|_2 \|\mathbf{s}_i\|_2}$, for the angle between the eigenscores $\widehat{\mathbf{s}}_i$ (see Methods) and the true local concordance $\mathbf{s}_i$ defined as

$$\mathbf{s}_i := ((\bar{\mathbf{P}}_{i\cdot}^{(1)})^\top \bar{\mathbf{P}}_{i\cdot}^*, (\bar{\mathbf{P}}_{i\cdot}^{(2)})^\top \bar{\mathbf{P}}_{i\cdot}^*, \ldots, (\bar{\mathbf{P}}_{i\cdot}^{(K)})^\top \bar{\mathbf{P}}_{i\cdot}^*) \in \mathbb{R}^K, \quad (1)$$

where $\bar{\mathbf{P}}_{i\cdot}^*$ is the $i$-th row of the normalized distance matrix $\bar{\mathbf{P}}^*$ for the underlying noiseless samples $\{\mathbf{Y}_i^*\}_{1 \le i \le n}$, defined as in (9) with $\mathbf{X}_i^{(k)}$'s replaced by $\mathbf{Y}_i^*$'s. Table 1 shows empirical mean and standard error (SE) of the averaged cosines $\frac{1}{n}\sum_{i=1}^n \cos \angle (\widehat{\mathbf{s}}_i, \mathbf{s}_i)$, over the family of datasets under the same low-dimensional structure associated with various $\theta$ as shown in Fig. 2a. Our simulations showed that $\cos \angle (\widehat{\mathbf{s}}_i, \mathbf{s}_i) \approx 1$, indicating that the eigenscores $\widehat{\mathbf{s}}_i$ essentially characterize the true concordance between the patterns contained in each candidate visualization and that of the underlying noiseless samples, evaluated locally with respect to sample $i$. This justifies the proposed eigenscore as a precise measure of performance of the candidate visualizations in preserving the underlying true signals.

To assess the quality of two meta-distance matrices, for each dataset, we compare the mean concordance $\frac{1}{n}\sum_{i=1}^n (\bar{\mathbf{P}}_{i\cdot}^{(k)})^\top \bar{\mathbf{P}}_{i\cdot}^*$ between

**Table 1 | Empirical mean and standard error (SE) of the averaged cosines $\frac{1}{n}\sum_{i=1}^{n} \cos \angle(\hat{\mathbf{s}}_i, \mathbf{s}_i)$, between the eigenscores and the true concordance measures, over each family of datasets associated with a given low-dimensional structure under various values of the SNR parameter $\theta$**

| Low-Dimensional Structure | Gaussian mixture | Smiley face | Mammoth |
|---|---|---|---|
| Simulation Setting | $(n, p) = (900, 500)$ | $(n, p) = (500, 300)$ | $(n, p) = (500, 300)$ |
| Empirical Mean (SE) | 0.992 ($10^{-5}$) | 0.986 ($10^{-5}$) | 0.990 ($10^{-5}$) |

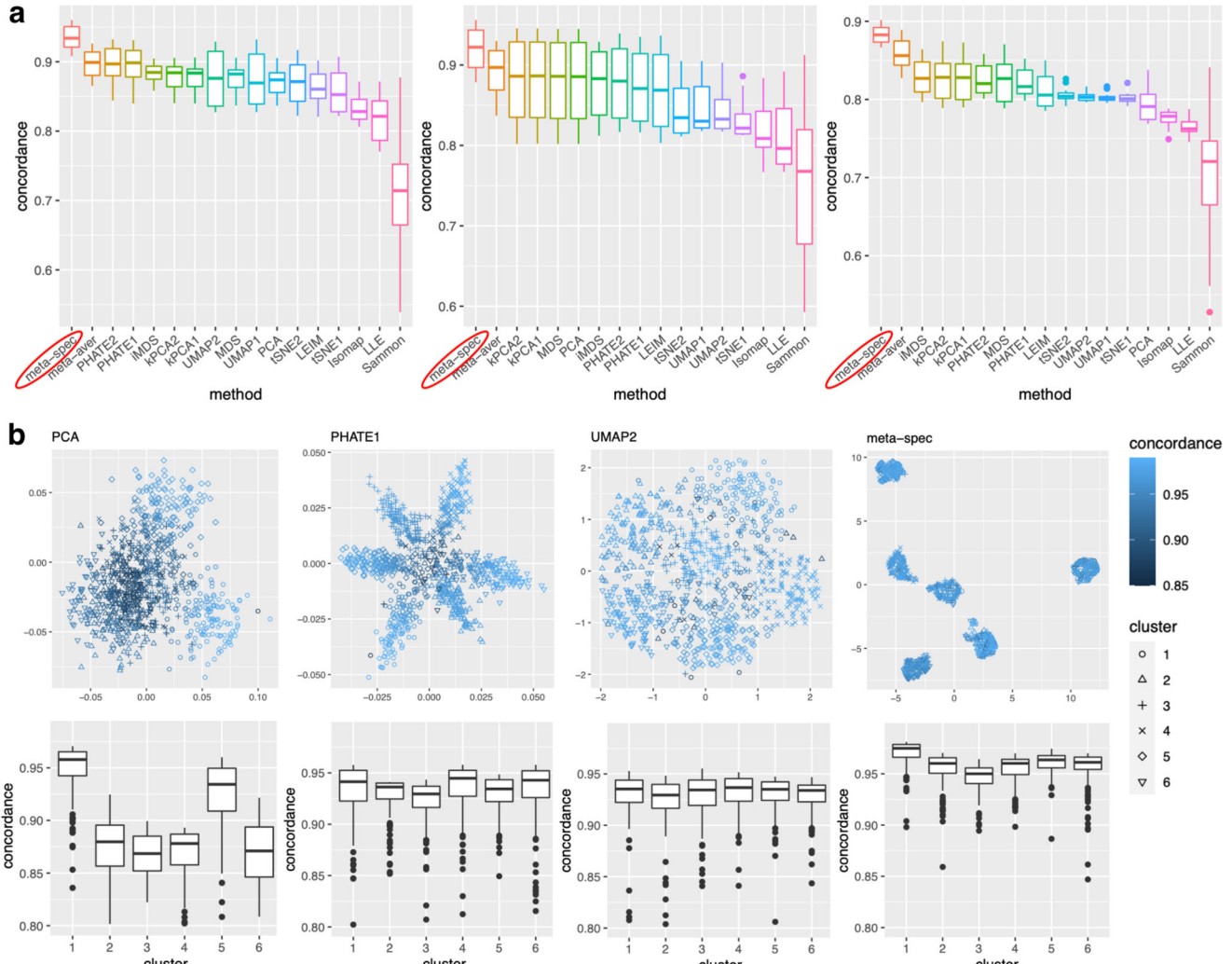

**Fig. 2 | Results from simulation studies. a** Boxplots (center line, median; box limits, upper and lower quartiles; points, outliers) of the mean concordance with the underlying true pattern for 15 candidate visualizations (HLLE omittted due to very low concordance) and the two meta-distance matrices under each simulation setting (Left: $n = 900$ independent samples generated from the Gaussian mixture model; Middle: $n = 500$ independent samples generated from the smiley face model; Right: $n = 500$ independent samples generated from the mammoth model) across various values of the SNR value $\theta$. See Supplementary Fig. 2 for

complete plots. **b** Results for $n = 900$ independent samples generated under the Gaussian mixture model. Top: examples of candidate visualizations along with their sample-wise concordance $\{(\bar{\mathbf{P}}_{i\cdot}^{(k)})^{\top}\bar{\mathbf{P}}_{i\cdot}^{*}\}_{1 \leq i \leq n}$ with the structure of noiseless samples, and the proposed meta-visualization using UMAP and the concordance $\{(\bar{\mathbf{P}}_{i\cdot}^{m})^{\top}\bar{\mathbf{P}}_{i\cdot}^{*}\}_{1 \leq i \leq n}$ for the proposed meta-distance. Bottom: boxplots (center line, median; box limits, upper and lower quartiles; points, outliers) of concordance measures as grouped by the true clusters. See Supplementary Fig. 3 for more examples.

---

the normalized distance of each candidate visualization and that of the underlying noiseless samples, and the mean concordance $\frac{1}{n}\sum_{i=1}^{n}(\bar{\mathbf{P}}_{i\cdot}^{m})^{\top}\bar{\mathbf{P}}_{i\cdot}^{*}$ between the obtained meta-distance and that of the underlying noiseless samples (see Methods). Figure 2a and Supplementary Fig. 2 show boxplots of these mean concordances for the 16 candidate visualizations and the two meta-distances under each setting of underlying structures across various values of $\theta$. We observe that for each of the three structures, our proposed meta-

distance is substantially more concordant with the underlying true patterns, than every candidate visualization and the naive meta-distance, indicating the superiority of the proposed meta-distance. To further demonstrate the advantage of the spectral meta-distance and its benefits to the final meta-visualization, we compared our proposed meta-visualization using UMAP, and candidate visualizations of a dataset under setting (i) with $\theta = 5$, and present their sample-wise concordance $\{(\bar{\mathbf{P}}_{i\cdot}^{(k)})^{\top}\bar{\mathbf{P}}_{i\cdot}^{*}\}_{1 \leq i \leq n}$ for each $k$, and $\{(\bar{\mathbf{P}}_{i\cdot}^{m})^{\top}\bar{\mathbf{P}}_{i\cdot}^{*}\}_{1 \leq i \leq n}$ of the

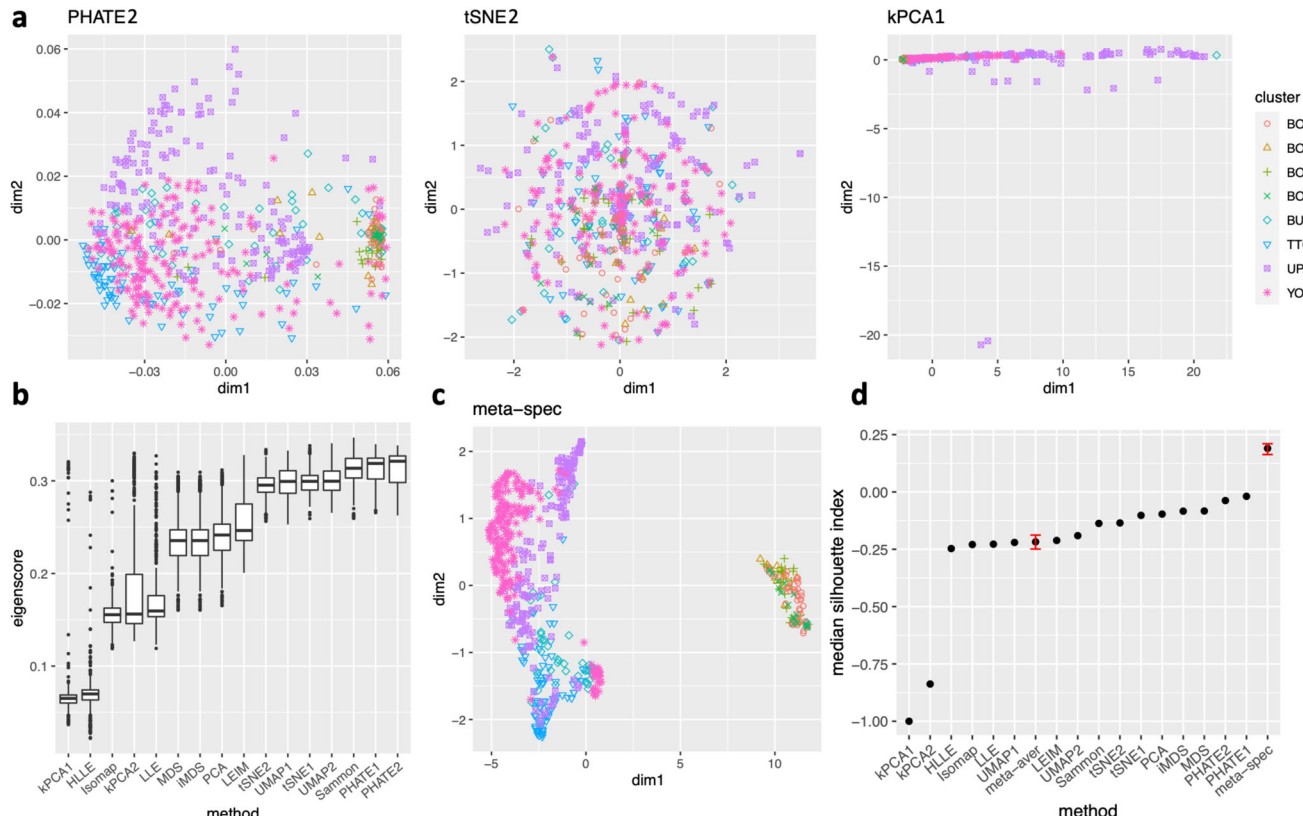

**Fig. 3 | Visualization of 590 fragments of texts from eight religious and biblical books. a** Three examples of candidate visualizations. The samples are marked by eight different symbols and colors according to their associated books. More examples are included in Supplementary Fig. 5. **b** Boxplots (center line, median; box limits, upper and lower quartiles; points, outliers) of eigenscores for all 16 candidate visualizations, each containing $n = 590$ samples. **c** The

proposed spectral meta-visualization using UMAP. **d** Median silhouette indices over $n = 590$ samples for the 16 candidate and 2 meta-visualizations. The error bars of the meta-visualizations indicate the variability (95% confidence interval over 50 rounds of repetitions) due to the visualization method (UMAP) applied to the meta-distance matrix (13). Source data are provided as a Source Data file.

proposed meta-distance (Fig. 2b and Supplementary Fig. 3). We observe that, while each individual method may capture some clusters in the dataset but misses others, the proposed meta-visualization is able to combine strengths of all the candidate visualizations in order to capture all the underlying clusters. Finally, to demonstrate the flexibility of our method with respect to higher intrinsic dimension $r$, under the setting (i), we further evaluated the performance of different methods for $r \in \{15, 30, 50\}$. Supplementary Fig. 4 shows consistent and superior performance of the proposed method compared to the other approaches.

## Visualizing clusters of religious texts

Cluster data are ubiquitous in scientific research and industrial applications. Our first real data example concerns $n = 590$ fragments of text, extracted from English translations of eight religious books or sacred scripts including Book of Proverb (BOP), Book of Ecclesiastes (BOE1), Book of Ecclesiasticus (BOE2), Book of Wisdom (BOW), Four Noble Truth of Buddhism (BUD), Tao Te Ching (TTC), Yogasutras (YOG) and Upanishads (UPA)[38]. All the text were pre-processed using natural language processing into a $590 \times 8265$ Document Term Matrix that counts frequency of 8265 atomic words, such as truth, diligent, sense, power, in each text fragment. In other words, each text fragment was treated as a bag of words, represented by a vector with 8265 features. The word counts were centred and normalized before downstream analysis.

As in our simulation studies, we still consider $K = 16$ candidate visualizations generated by 12 different methods with various tuning parameters (see Supplementary file Section A.2 for details). Figure 3a

contains examples of candidate visualizations obtained by PHATE, t-SNE, and kPCA, whose median eigenscores were ranked top, middle and bottom among all the visualizations (Fig. 3b), respectively. More examples are included in Supplementary Fig. 5. In each visualization, the samples (text fragments) were colored by their associated books, showing how well the visualization captures the underlying clusters of the samples. The usefulness and validity of the eigenscores in Fig. 3b can be verified empirically, by visually comparing the clarity of cluster patterns demonstrated by each candidate visualizations in Fig. 3a and in Supplementary Fig. 5. Figure 3c is the proposed meta-visualization (hereafter we used meta-spec and meta-aver to refer to the final meta-visualizations rather than the meta-distance matrices as in Section 2) of the samples by applying UMAP to the meta-distance matrix, which shows substantially better clustering of the text fragments in accordance with their sources. In addition, the meta-visualization also reflected deeper relationship between the eight religious books, such as the similarity between the two Hinduism books YOG and UPA, the similarity between Buddhism (BUD) and Taoism (TTC), the similarity between the four Christian books BOE1, BOE2, BOP, and BOW, as well as the general discrepancy between Asian religions (Hinduism, Buddhism, Taoism) and non-Asian religions (Christianity). All of these important phenomena, while salient in our meta-visualization, only appeared vaguely in very few candidate visualizations such as those produced by PHATE (Fig. 3a) and UMAP (Supplementary Fig. 5).

To quantitatively evaluate the preservation of the underlying clustering pattern, we computed for each visualization the Silhouette indices[39] with respect to the underlying true cluster membership,

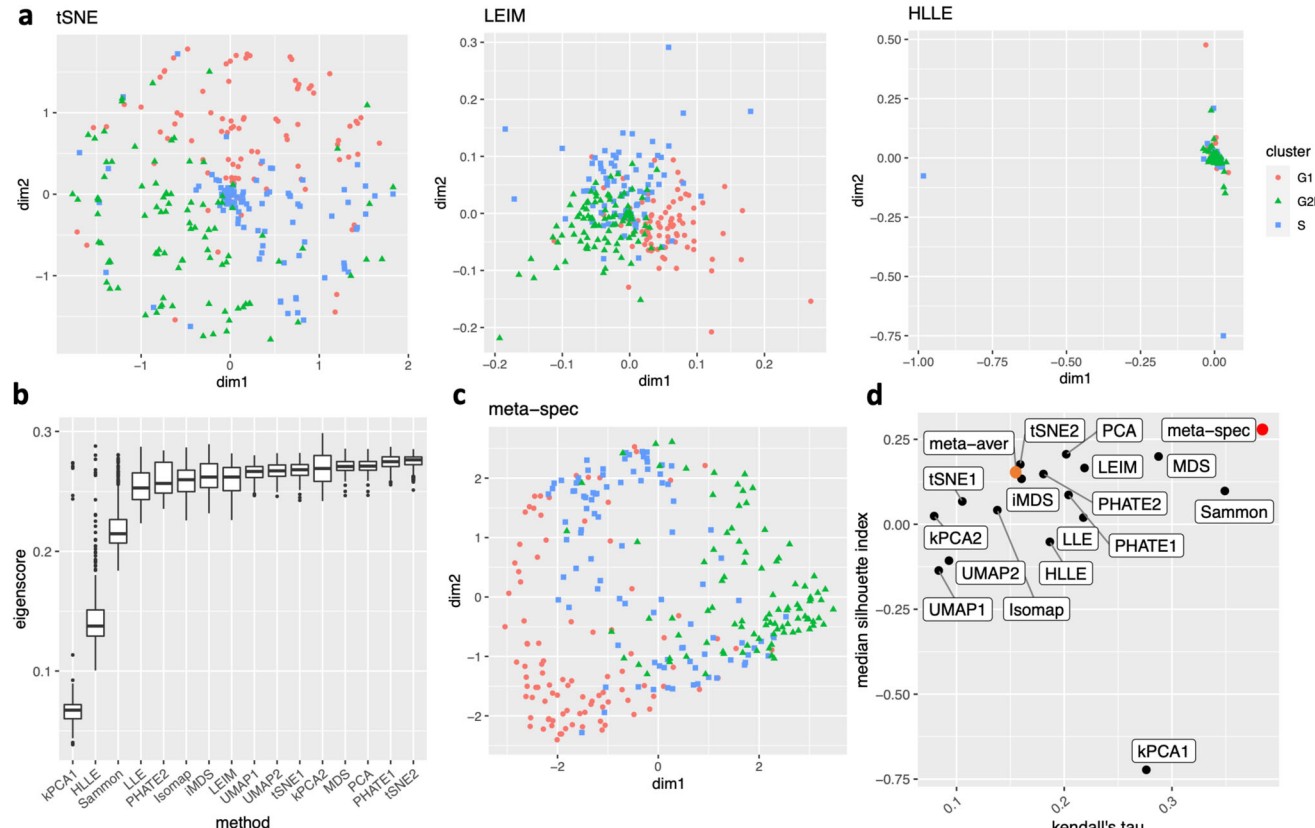

**Fig. 4 | Visualization of the cell cycle of 288 mouse emryonic stem cells. a** Three examples of candidate visualizations. The cells are marked by three different symbols and colors according to their associated cell cycle stages. More examples are included in Supplementary Fig. 16. **b** Boxplots (center line, median; box limits, upper and lower quartiles; points, outliers) of eigenscores for all 16 candidate visualizations, each containing $n = 288$ samples. **c** The proposed meta-visualization using kPCA. **d** Median Silhouette indices versus Kendall's tau statistics for the 16 candidate and the 2 meta-visualizations (Red: proposed spectral meta-visualization; Orange: naive simple average meta-visualization). For both metrics, a higher value indicates a better visualization of the respective structure (cluster/cycle). Source data are provided as a Source Data file.

based on the normalized pairwise-distance matrices of the embeddings defined in (9). The Silhouette index (see Supplementary file Section A.2 for its definition), defined for each individual sample in a visualization, measures the amount of discrepancy between the within-class distances and the inter-class distances with respect to a given sample. As a result, for a given visualization, its Silhouette indices altogether indicate how well the underlying cluster pattern is preserved in a visualization, and higher Silhouette indices indicate that the underlying clusters are more separate. Empirically, we observed a notable correlation ($\rho = 0.679$) between the median Silhouette indices and the median eigenscores across the candidate visualizations (Supplementary Fig. 6). In addition, for each candidate visualization, we found that samples with higher Silhouette index tend to have higher eigenscores (Supplementary Fig. 7), demonstrating the effectiveness of eigenscores, and its benefits on the final meta-visualization. In Fig. 3d, we show that, even taking into account the stochasticity of the visualization method (UMAP) applied to the meta-distance matrix, our meta-visualization had the median Silhouette index much higher than those of the candidate visualizations, as well as that of the meta-visualization based on the naive meta-distance (meta-aver). It is of interest to note that meta-spec was the only visualization with a positive median Silhouette index, showing its better separation of clusters compared with other visualizations. Importantly, the proposed meta-visualization was not sensitive to the specific visualization method applied to the meta-distance matrices – similar results were obtained when we replaced UMAP by PHATE, the method having the highest median eigenscore in Fig. 3c, or t-SNE, for meta-visualization (Supplementary Fig. 6).

## Visualizing cell cycles

Our second real data example concerns visualization of a different low-dimensional structure, namely, a mixture of cycle and clusters, contained in the gene expression profile of a collection of mouse embryonic stem cells, as a result of the cell cycle mechanism. The cell cycle, or cell-division cycle, is the series of events that take place in a cell that cause it to divide into two daughter cells. Identifying the cell cycle stages of individual cells analyzed during development is important for understanding its wide-ranging effects on cellular physiology and gene expression profiles. Specifically, we consider a dataset containing $n = 288$ mouse embryonic stem cells[40], whose underlying cell cycle stages were determined using flow cytometry sorting. Among them, one-third (96) of the cells are in the G1 stage, one-third in the S stage, and the rest in the G2M stage. The raw count data were preprocessed and normalized, leading to a dataset consisting of standardized expression levels of 1147 cell-cycle-related genes for the 288 cells (Methods).

We obtained 16 candidate visualizations as before, and applied our proposed method. Figure 4a contains examples of candidate visualizations obtained by t-SNE, LEIM, and kPCA, whose median eigenscores were ranked top, middle and bottom among all the visualizations, respectively, and the cells were colored according to their true cell cycle stages. Figure 4b contains the boxplots of eigenscores for the candidate visualizations, indicating the overall quality of each visualization. The variation of eigenscores within each candidate visualization suggests that different visualizations have their own unique features and strengths to be contributed to the meta-visualization (Supplementary Fig. 8). Figure 4c is the proposed meta-

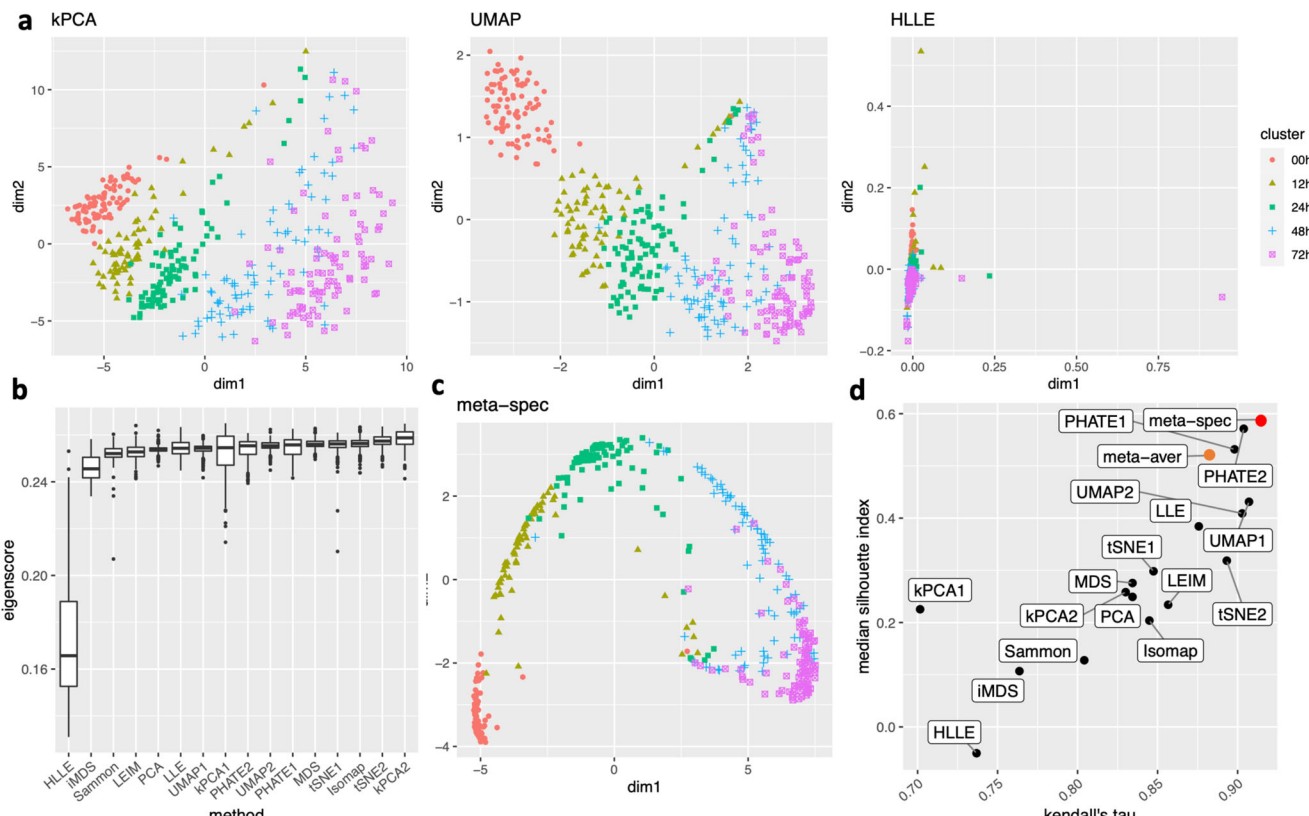

**Fig. 5 | Visualization of 421 cells undergoing differentiation. a** Three examples of candidate visualizations. The individual cells are marked by five different symbols and colors according to the time points they were captured and sequenced. More examples are included in Supplementary Fig. 11. **b** Boxplots (center line, median; box limits, upper and lower quartiles; points, outliers) of eigenscores for all 16 candidate visualizations, each containing *n* = 421 samples. **c** The proposed meta-visualization using kPCA. **d** Median Silhouette indices versus Kendall's tau statistics for the 16 candidate and the 2 meta-visualizations (Red: proposed spectral meta-visualization; Orange: naive simple average meta-visualization). Source data are provided as a Source Data file.

visualization by applying kPCA to the meta-distance matrix. Comparing with Fig. 4a, the proposed meta-visualization showed better clustering of the cells according to their cell cycle stages, as well as a more salient cyclic structure underlying the three cell cycle stages (Supplementary Figure 9). To quantify the performance of each visualization in terms of these two underlying structures (cluster and cycle), we considered two distinct metrics, namely, the median Silhouette index with respect to the underlying true cell cycle stages, and the Kendall's tau statistic[41] between the inferred relative order of the cells and their true orders on the cycle. Specifically, to infer the relative order of cells, we projected the coordinates of each visualization to the two-dimensional unit circle centred at the origin (Supplementary Fig. 9), and then determined the relative orders based on the cells' respective projected positions on the unit circle. Figure 4d shows that the proposed meta-visualization was significantly better than all the candidate visualizations and the naive meta-visualization in representing both aspects of the data.

**Visualizing trajectories of cell differentiation**
Our third real data example concerns visualization of a mixed pattern of a trajectory and clusters underlying the gene expression profiles of a collection of cells undergoing differentiation[42]. Specifically, 421 mouse embryonic stem cells were induced to differentiate into primitive endoderm cells. After the induction of differentiation, the cells were dissociated and individually captured at 12- or 24-hour intervals (0, 12, 24, 48 and 72 h), and each cell was sequenced to obtain the final total RNA sequencing reads using the random displacement amplification sequencing technology. As a result, at each of the five-time points, there were about 70 to 90 cells captured and sequenced. The raw

count data were preprocessed and normalized (Methods), leading to a dataset consisting of standardized expression levels of 500 most variable genes for the 421 cells.

Again, we obtained 16 candidate visualizations as before, and applied our proposed method. In Fig. 5a–c we show examples of candidate visualizations, boxplots of the eigenscores, and the meta-visualization using kPCA. The global (Fig. 5b) and local (Supplementary Fig. 10) variation of eigenscores demonstrated contribution of different visualizations to the final meta-visualization according to their respective performance. We observed that some candidate visualizations such as kPCA, UMAP (Fig. 5a) and PHATE (Supplementary Fig. 11) to some extent captured the underlying trajectory structure consistent with the time course of the cells. However, the meta-visualization in Fig. 5c showed much more salient patterns in terms of both the underlying trajectory and the cluster pattern among the cells, by locally combining strengths of the individual visualizations (Supplementary Fig. 10). We quantified the performance of visualizations from these two aspects using the median Silhouette index with respect to the underlying true cluster membership (i.e., batches of time course) and Kendall's tau statistic between the inferred cell order and the true order along the progression path. To infer the relative order of the cells from a visualization, we ordered all the cells based on the two-dimensional embedding along the direction that explained the most variability of the cells. In Fig. 5d, we observed that, the proposed meta-visualization had the largest median Silhouette index as well as the largest Kendall's tau statistic, compared with all the candidate visualizations and the naive meta-visualization, showing the superiority of the proposed meta-visualization in both aspects.

## Computational cost

For datasets of moderate size as the ones analyzed in the previous sections, the proposed method had a computational cost comparable to that of t-SNE or UMAP for generating a single candidate visualization (Supplementary Fig. 12). As for very large and high-dimensional datasets, there are a few features of the proposed algorithm that make it readily scalable. First, although our method relies on computing the leading eigenvector of generally non-sparse matrices, these matrices (i.e., $\mathbf{G}_i$ in Algorithm 1) are of dimension $K \times K$, where $K$ – the number of candidate visualizations – is usually much smaller compared to the sample size $n$ or dimensionality $p$ of the original data. Thus, for each sample $i$, the computational cost due to the eigendecomposition is mild. Second, given the candidate visualizations, our proposed algorithm is independent of the dimensionality ($p$) of the original dataset, as it only requires as input a set of low-dimensional embeddings produced by different visualization methods. Third, since our algorithm computes the eigenscores and the meta-distance with respect to each sample individually, the algorithm can be easily parallelized and carried out in multiple cores to further reduce time cost.

To demonstrate the computational efficiency of the proposed method for large and high-dimensional datasets, we evaluated the proposed method on real single-cell transcriptomic datasets[43] of various sample sizes ($n \in \{1000, 2000, 4000, 8000, 14000\}$ cells of nine different cell types from the neurogenic regions of mice) and dimensions ($p \in \{500, 1000, 2000\}$ genes). For each dataset, we obtained 11 candidate visualizations and applied Algorithm 1 to generate the final meta-visualization (Methods). Supplementary Fig. 13b contains boxplots of median Silhouette indices for each candidate visualizations and the meta-visualization (highlighted in red) with respect to the underlying true cell types, showing the stable and superior performance of the proposed method under various sample sizes and dimensions. In Supplementary Fig. 13a, we compared the running time for generating the 11 candidate visualizations, and that for generating the meta-visualizations based on Algorithm 1, on a MacBook Pro with 2.2 GHz 6-Core Intel Core i7. In general, as $n$ became large, the running time of the proposed algorithm also increased, but remained much less than that for generating the candidate visualizations. The difference in time cost became more significant as $n$ increased, demonstrating that for very large and high-dimensional datasets the computational cost essentially comes from generating candidate visualizations, rather than from the meta-visualization step. In particular, for dataset of sample size as large as 8000 and of dimension 2000, it took about 60 mins to generate all the 11 candidate visualizations, and took about additional 12 mins to generate the meta-visualization. Moreover, Supplementary Fig. 13a also demonstrated that, for each $n$, when $p$ increased, the running time for generating the candidate visualizations was longer, but the time cost for meta-visualization remained about the same (difference less than one minute). We also note that users often create multiple visualizations for data exploration, and our approach can simply reuse these visualizations with little additional computational cost.

## Theoretical guarantees

We develop a general and flexible theoretical framework, to investigate the statistical properties of the proposed methods, as well as the fundamental principles behind its empirical success. Throughout, for a matrix $\mathbf{A} = (a_{ij}) \in \mathbb{R}^{n \times n}$, we define its spectral norm as $\|\mathbf{A}\| = \sup_{\|\mathbf{x}\|_2 \leq 1} \|\mathbf{A}\mathbf{x}\|_2$. For sequences $\{a_n\}$ and $\{b_n\}$, we write $a_n = o(b_n)$ or $b_n \gg a_n$ if $\lim_n a_n/b_n = 0$, and write $a_n \asymp b_n$ if there exists constants $C_1, C_2 > 0$ such that $C_1 b_n \leq a_n \leq C_2 b_n$ for all $n$. We consider visualizing a $p$-dimensional dataset $\{\mathbf{Y}_i\}_{1 \leq i \leq n}$ containing $n$ samples. From $\{\mathbf{Y}_i\}_{1 \leq i \leq n}$, suppose we obtain a collection of $K$ (candidate) visualizations of the data, produced by various visualization methods.

We denote these visualizations as two-dimensional embeddings $\{\mathbf{X}_i^{(k)}\}_{1 \leq i \leq n} \subset \mathbb{R}^2$ for $k \in \{1, 2, \ldots, K\}$. As can be seen from Methods Section, there are two key ingredients of our proposed method, namely, the eigenscores $\{\hat{s}_i\}_{1 \leq i \leq n}$ for evaluating the candidate visualizations, and the meta-distance matrix $\bar{\mathbf{P}}^m$ that combines multiple candidate visualizations to obtain a meta-visualization. To formally study their properties, we introduce a generic model for the collection of $K$ candidate visualizations produced by multiple visualization methods, with possibly different settings of tuning parameters for a single method as considered in previous sections. Specifically, we assume $\{\mathbf{Y}_i\}_{1 \leq i \leq n}$ are generated as

$$\mathbf{Y}_i = \mathbf{Y}_i^* + \mathbf{Z}_i, \qquad i = 1, 2, \ldots, n, \qquad (2)$$

where $\{\mathbf{Y}_i^*\}_{1 \leq i \leq n}$ are the underlying noiseless samples and $\{\mathbf{Z}_i\}_{1 \leq i \leq n}$ are the random noises. Recall that $\{\mathbf{P}^{(k)}\}_{1 \leq k \leq K}$ are the distance matrices associated to the candidate visualizations (see Methods). Then, for the candidate visualizations, we consider a scaled signal-plus-noise expression

$$\mathbf{P}_{i.}^{(k)} = c_{i,k}(\mathbf{P}_{i.}^* + \mathbf{h}_i^{(k)}), \qquad k = 1, 2, \ldots, K, \qquad (3)$$

induced by (2), where $c_{i,k} \geq 0$ is a global scaling parameter, $\mathbf{P}_{i.}^*$ is the $i$-th row of the pairwise distance matrix $\mathbf{P}^* = (\| \mathbf{Y}_i^* - \mathbf{Y}_j^* \|_2)_{1 \leq i, j \leq n}$ of the underlying noiseless samples, and $\mathbf{h}_i^{(k)}$ is a random vector characterizing the relative distortion of $\mathbf{P}_{i.}^{(k)}$ associated to the $k$-th candidate visualization, from the underlying true pattern $\mathbf{P}_{i.}^*$. Before characterizing the distributions of $\{\mathbf{h}_i^{(k)}\}_{1 \leq k \leq K}$, we point out that, in principle, the relative distortions $\{\mathbf{h}_i^{(k)}\}_{1 \leq k \leq K}$ are jointly determined by the random noises $\{\mathbf{Z}_i\}_{1 \leq i \leq n}$ in (2), and the features and relations between of the specific visualization methods. Importantly, in line with what is often encountered in practice, equation (3) allows for flexible and possibly distinct scaling and directionality for different candidate visualizations, by introducing the visualization-specific parameter $c_k$, and by focusing on the pairwise distance matrices, rather than the low-dimensional embeddings $\{\mathbf{X}_i^{(k)}\}_{1 \leq i \leq n}$ themselves.

To quantitatively describe the variability of the distortions $\{\mathbf{h}_i^{(k)}\}_{1 \leq k \leq K}$ across $K$ candidate visualizations, we assume

(C1a) $\{\mathbf{h}_i^{(k)}\}_{1 \leq k \leq K}$ are identically distributed sub-Gaussian vectors with parameter $\sigma^2$, that is, for any deterministic unit vector $\mathbf{g} \in \mathbb{R}^n$, we have $\mathbb{E} \exp\{(\mathbf{h}_i^{(k)})^\top \mathbf{g}\} \leq \exp(\sigma^2/2)$, and that $\| \mathbf{h}_i^{(k)} \|_2^2 = c\sigma^2 n(1 + o(1))$ for some constant $c > 0$ with high probability, that is, with probability at least $1 - n^{-D}$ for some large constant $D > 0$ for all sufficiently large $n$.

This assumption makes (3) a generative model for $\{\mathbf{P}_{i.}^{(k)}\}_{1 \leq k \leq K}$ with ground truth $\mathbf{P}_{i.}^*$ and random distortions, where the variance parameter $\sigma$ describes the average level of the distortions of candidate visualizations from the truth after proper scaling. In relation to (2), such a condition can be satisfied when the signal structure $\{\mathbf{Y}_i^*\}_{1 \leq i \leq n}$ is finite, the noise $\{\mathbf{Z}_i\}_{1 \leq i \leq n}$ is sub-Gaussian, and the dimension reduction map underlying the candidate visualization is bounded and sufficiently smooth. See Supplementary file Section B.2 for details. In addition, we also need to characterize the correlations among these random distortions, not only because the candidate visualizations are typically obtained from the same dataset $\{\mathbf{Y}_i\}_{1 \leq i \leq n}$, but also because of the possible similarity between the adopted visualization methods, such as MDS and iMDS, or t-SNE under different tuning parameters. Specifically, for any $j, k \in \{1, 2, \ldots, K\}$, we define the cross-visualization covariance $\mathbf{\Sigma}_{jk} = \mathbb{E}\mathbf{h}_i^{(j)}(\mathbf{h}_i^{(k)})^\top$, and quantify the level of dependence between a pair of candidate visualizations by $\rho_{jk} = \|\mathbf{\Sigma}_{jk}\|/\sigma^2$. By Condition (C1a), we have $\rho_{jj} \leq 1$ for all $j$. For all correlation parameters $\{\rho_{jk}\}_{1 \leq j, k \leq K}$, we assume

(C1b) The matrix $\mathbf{R} = (\rho_{jk})_{1 \le j,k \le K}$ satisfies $\rho := \|\mathbf{R}\| = o(K)$.

Condition (C1b) covers a wide range of correlation structures among the candidate visualizations, allowing in particular for a subset of highly correlated visualizations possibly produced by very similar methods. The parameter $\rho$ characterizes the overall correlation strength among the candidate visualizations, which is assumed to be not too large. As a comparison, note that a set of pairwise independent candidate visualizations implies that $\rho \approx 1$, whereas a set of identical candidate visualizations have $\rho \approx K$. In particular, the requirement $\rho = o(K)$ can be satisfied if, for example, among $K$ candidate visualizations, there are subsets of at most $\sqrt{K}$ visualizations that are produced by very similar procedures, such as by the same method under different tuning parameters, so that $\rho \le \sqrt{K} = o(K)$. When Condition (C1b) fails, as all the candidate visualizations are essentially similarly distorted from truth, combination of them will not be substantially more informative than each individual visualization.

Under Condition (C1a), it holds that $\mathbb{E} \| \mathbf{h}_i^{(k)} \|_2 \asymp \sigma\sqrt{n}$. Hence, we can use the quantity $\frac{\|\mathbf{P}_{i\cdot}^*\|_2}{\sigma\sqrt{n}}$ to characterize the overall SNR in the candidate visualizations as modelled by (3), which reflects the average quality of the candidate visualizations in preserving the underlying true patterns around sample $i$. Before stating our main theorems, we first introduce our main assumption on the minimal SNR requirement, that is,

(C2) For $(\sigma, \rho)$ defined in (C1a) and (C1b), it holds that $\frac{\|\mathbf{P}_{i\cdot}^*\|_2}{\sigma\sqrt{n}} \gg \sqrt{\rho/K}$ and $K = o(n)$ as $n \to \infty$.

Our algorithm is expected to perform well if $\sqrt{\rho/K}$ is small relative to the overall SNR. The condition $K = o(n)$ is easily satisfied for a sufficiently large dataset.

Recall that $\mathbf{s}_i = ((\mathbf{P}_{i\cdot}^{(1)})^\top \bar{\mathbf{P}}_{i\cdot}^*, (\mathbf{P}_{i\cdot}^{(2)})^\top \bar{\mathbf{P}}_{i\cdot}^*, \ldots, (\mathbf{P}_{i\cdot}^{(K)})^\top \bar{\mathbf{P}}_{i\cdot}^*)$. The following theorem concerns the convergence of eigenscores to the true concordance $\mathbf{s}_i$, and is proved in Supplementary file Section B.3.

**Theorem 1.** Under Conditions (C1a) (C1b) and (C2), for each $i \in \{1, 2, \ldots, n\}$, it holds that $\cos \angle(\widehat{\mathbf{s}}_i, \mathbf{s}_i) = \frac{(\widehat{\mathbf{s}}_i)^\top \mathbf{s}_i}{\|\widehat{\mathbf{s}}_i\|_2 \|\mathbf{s}_i\|_2} \to 1$ in probability as $n \to \infty$.

Theorem 1 implies that, as long as the candidate visualizations contain sufficient amount of information about the underlying true structure, and are not terribly correlated, the proposed eigenscores $\{\widehat{\mathbf{s}}_i\}_{1 \le i \le n}$ are quantitatively reliable, as they converge to the actual quality measures $\{\mathbf{s}_i\}_{1 \le i \le n}$ asymptotically. In other words, the eigenscores provide a point-wise consistent estimation of the concordance between the candidate visualizations as summarized by $\{\mathbf{P}^{(k)}\}_{1 \le k \le K}$ and the underlying true patterns $\mathbf{P}^*$, justifying the empirical observations in Table 1. Importantly, Condition (C2) suggests that our proposed eigenscores may benefit from a larger number $K$ of candidate visualizations, or a smaller overall correlation $\rho$, that is, a collection of functionally more diverse candidate visualizations.

Our second theorem concerns the guaranteed performance of our proposed meta-distance matrix and its improvement upon the individual candidate visualizations in the large-sample limit.

**Theorem 2.** Under Conditions (C1a) (C1b) and (C2), for each $i \in \{1, 2, \ldots, n\}$, it holds that $\cos \angle(\bar{\mathbf{P}}_{i\cdot}^m, \mathbf{P}_{i\cdot}^*) \to 1$ in probability as $n \to \infty$. Moreover, for any constant $\delta \in (0, 1)$, there exist a constant $C > 0$ such that, whenever $\| \mathbf{P}_{i\cdot}^* \|_2 \le C\sigma\sqrt{n}$, we have $\max_{1 \le k \le K} \cos \angle(\mathbf{P}_{i\cdot}^{(k)}, \mathbf{P}_{i\cdot}^*) < 1 - \delta$ in probability as $n \to \infty$.

Theorem 2 is proved in Supplementary file Section B.4. In addition to the point-wise consistency of $\bar{\mathbf{P}}^m$ as described by $\cos \angle(\bar{\mathbf{P}}_{i\cdot}^m, \mathbf{P}_{i\cdot}^*) \to 1$ in probability, Theorem 2 also ensures that the proposed meta-distance is in general no worse than the individual candidate

visualizations, suggesting a competitive performance of the meta-visualization. In particular, if in addition to Conditions (C1a) (C1b) and (C2) we also have $\frac{\|\mathbf{P}_{i\cdot}^*\|_2}{\sigma\sqrt{n}} \le C$, that is, the magnitude of the random distortions from the true structure $\mathbf{P}_{i\cdot}^*$ is relatively large, then each candidate visualization necessarily has at most mediocre performance, i.e., $\max_{1 \le k \le K} \cos \angle(\mathbf{P}_{i\cdot}^{(k)}, \mathbf{P}_{i\cdot}^*) < 1 - \delta$ in probability. In such cases, the proposed meta-distances is still consistent and thus strictly better than all candidate visualizations. Theorem 2 justifies the superior performance of the spectral meta-visualization demonstrated in previous sections, compared with 16 candidate visualizations.

Among the three conditions required for the consistency of the proposed meta-distance matrix, Condition (C2) is most critical as it describes the minimal SNR requirement, that is, how much information the candidate visualizations altogether should contain about the underlying true structure of the data. In this connection, our theoretical analysis indicates that, in fact, such a signal strength condition is also necessary, not only for the proposed method, but for any possible methods. More specifically, in Supplementary file Section B.6, we proved (Theorem 4) that, it's impossible to construct a meta-distance matrix that is consistent when Condition (C2) is violated. This result shows that the settings where our meta-visualization algorithm works well is essentially the most general setting possible.

## Robustness of spectral weighting against adversarial visualizations

In our numerical studies, in addition to the proposed meta-visualization, we also considered the meta-visualization based on the naive meta-distance matrix $\bar{\mathbf{P}}^a$, whose rows are

$$\bar{\mathbf{P}}_{i\cdot}^a = \frac{1}{K} \sum_{k=1}^{K} \bar{\mathbf{P}}_{i\cdot}^{(k)} \in \mathbb{R}^n, \tag{4}$$

which is a simple average across all the candidate visualizations. We observed in all our real-world data analyses that, such a naive meta-visualization only had mediocre performance compared to the candidate visualizations (Figs. 3, 4, and 5), much worse than the proposed spectral meta-visualization. The empirical observations suggest the advantage of informative weighting for combining candidate visualizations.

The empirically observed suboptimality of the non-informative weighting procedure can justified rigorously by theory. Our next theorem concerns the behavior of the proposed meta-distance matrix $\bar{\mathbf{P}}^m$ and the naive meta-distance matrix $\bar{\mathbf{P}}^a$ when combining a mixture of well-conditioned candidate visualizations, as characterized by our assumptions (C1a) (C1b) and (C2), and some adversarial candidate visualizations whose pairwise-distance matrices does not contain any information about the true structure. Specifically, we suppose among all the $K$ candidate visualizations, there is a collection $\mathcal{C}_0$ of $(1 - \eta)K$ well-conditioned candidate visualizations for some small $\eta \in (0, 1)$, and a collection $\mathcal{C}_1$ of $\eta K$ adversarial candidate visualizations.

**Theorem 3.** For any $i \in \{1, 2, \ldots, n\}$, suppose among all the $K$ candidate visualizations, there is a collection $\mathcal{C}_0$ of $(1 - \eta)K$ candidate visualizations for some small $\eta \in (0, 1)$ satisfying Conditions (C1a) (C1b) and (C2), and a collection $\mathcal{C}_1$ of $\eta K$ adversarial candidate visualizations such that $(\mathbf{P}_{i\cdot}^{(k)})^\top \mathbf{P}_{i\cdot}^* = 0$ for all $k \in \mathcal{C}_1$. Then, for the proposed meta-distance $\bar{\mathbf{P}}^m$, we still have $\cos \angle(\bar{\mathbf{P}}_{i\cdot}^m, \mathbf{P}_{i\cdot}^*) \to 1$ in probability as $n \to \infty$. However, for the naive meta-distance $\bar{\mathbf{P}}^a$, even if $\| \mathbf{P}_{i\cdot}^* \|_2 \gg \sigma\sqrt{n}$, we have $\cos \angle(\bar{\mathbf{P}}_{i\cdot}^a, \mathbf{P}_{i\cdot}^*) < 1 - \eta$ in probability as $n \to \infty$.

Theorem 3 is proved in Supplementary file Section B.5. By Theorem 3, on the one hand, even when there are a small portion of really

poor (adversarial) candidate visualizations to be combined with other relatively good visualizations, the proposed method still perform well thanks to the consistent eigenscore weighting in light of Theorem 1. On the other hand, no matter how strong the SNR is for those well-conditioned candidate visualizations, the method based on non-informative weighting is strictly sub-optimal. Indeed, when $\| \mathbf{P}_{i.}^{*} \|_2 \gg \sigma\sqrt{n} \asymp \mathbb{E} \| \mathbf{h}_i \|_2$, although we have $\cos\angle(\mathbf{P}_{i.}^{(k)}, \mathbf{P}_{i.}^{*}) \to 1$ in probability for all $k \in \mathcal{C}_0$, the non-informative weighting would suffer from the non-negligible negative effects from the adversarial visualizations in $\mathcal{C}_1$, causing a strict deviation from $\mathbf{P}^{*}$; see, for example, Figs. 3–5 (b)(d) for empirical evidences from real-world data.

### Limitations of original noisy high-dimensional data

The proposed eigenscores provide an efficient and consistent way of evaluating the performance of the candidate visualizations. As mentioned in Introduction, a number of metrics have been proposed to quantify the distortion of a visualization by comparing the low-dimensional embedding directly with the original high-dimensional data. Such metrics essentially treat the original high-dimensional data as the ground truth, and do not take into account the noisiness of the high-dimensional data. However, for many datasets arising from real-world applications, the observed datasets, as modelled by (2), are themselves very noisy, which may not make an ideal reference point for evaluating a visualization that probably has already significantly denoised the data through dimension reduction. For example, all the three real-world datasets we have considered contain much more features than number of samples. In each case, there are some underlying clusters among the samples, but the original datasets showed significantly weaker cluster structure compared to most of the 16 candidate visualizations (Supplementary Figure 14), suggesting that directly comparing a visualization with the noisy high-dimensional data may be misleading. In this respect, our theorems indicate that the proposed spectral method is able to precisely assess and effectively combine multiple visualizations to better grasp the underlying noiseless structure $\mathbf{P}^{*}$, without referring to the original noisy datasets, making it more robust, flexible, and computationally more efficient.

### Benefits of including more functionally diverse visualizations

Our theoretical analysis implies that the proposed meta-visualization may benefit from a large number (larger $K$) of functionally diverse (small $\rho$) candidate visualizations. To empirically verify this theoretical observation, we focused on the religious and biblical text data and the mouse embryonic stem cells data, and obtained spectral meta-visualizations based on a smaller but relatively diverse collection of 5 candidate visualizations, produced by arguably the most popular methods, namely, t-SNE, PHATE, UMAP, PCA and MDS, respectively. Compared with the 16 candidate visualizations considered earlier, here we have presumably similar $\rho$ but much smaller $K$. As a result, for the religious and biblical texts data, the meta-visualization had a median Silhouette index 0.187 (Supplementary Figure 15), which was smaller than the median Silhouette index 0.275 based on the 16 candidate visualizations as in Fig. 3d; for the cell cycle data, the meta-visualization had a median Silhouette index -0.062 and a Kendall's tau statistic 0.313, both smaller than the respective values based on the 16 candidate visualizations as in Fig. 4d. On the other hand, we also evaluated the effect when $\rho$ is increased but $K$ remains fixed. Specifically, we obtained 16 candidate visualizations, all produced by PHATE with varying nearest neighbor parameters, the final spectral meta-visualization had a median Silhouette index 0.094, which was even lower than the above meta-visualization based on five distinct methods, although being still slightly better than the 16 PHATE-based candidate visualizations (Supplementary Figure 15). These empirical evidences were in line with our theoretical predictions, suggesting benefits of including more diverse visualizations.

## Discussion

We developed a spectral method in the current study to assess and combine multiple data visualizations. The proposed meta-visualization combines candidate visualizations through an arithmetic weighted average of their normalized distance matrices, by their corresponding eigenscores. Although the proposed method was shown both in theory and numerically to outperform the individual candidate visualizations and their naive combination, it is still unclear whether there exists any other forms of combinations that lead to even better meta-visualizations. For example, one could consider constructing a meta-distance matrix using the geometric or harmonic (weighted) average, or an average based on barycentric coordinates[44]. We plan to investigate such problems concerning how to optimally combining multiple visualizations in a subsequent work.

Although originally developed for data visualization, the proposed method can be useful for other supervised and unsupervised machine learning tasks, such as combining multiple algorithms for clustering, classification, or prediction. For example, for a given dataset, if one has a collection of predicted cluster memberships produced by multiple clustering algorithms, one could construct cluster membership matrices with $(i,j)$-th entry being 0 if sample $i$ and $j$ are not assigned to the same cluster and being 1 otherwise. Then we may define the similarity matrix as in (11), obtain the eigenscores for the candidate clusterings, and a meta-clustering using (13). It is of interest to know its empirical performance and if the fundamental principles unveiled in the current work continue to hold for such broader range of learning tasks.

## Methods
### Eigenscore and meta-visualization methodology

Throughout, without loss of generality, we assume that for visualization purpose the target embedding is two-dimensional, although our discussion applies to any finite-dimensional embedding.

**Algorithm 1.** Spectral assessment and combination of multiple data visualizations

Input: candidate visualizations $\{\mathbf{X}_i^{(k)}\}_{1 \le i \le n}$ for $k \in \{1, 2, \dots, K\}$.

1. Construct normalized pairwise-distance matrices: for each $k \in \{1, 2, \dots, K\}$, calculate

$$\bar{\mathbf{P}}^{(k)} = [\mathbf{D}^{(k)}]^{-1} \mathbf{P}^{(k)}, \qquad (5)$$

where $\mathbf{P}^{(k)} = (\| \mathbf{X}_i^{(k)} - \mathbf{X}_j^{(k)} \|_2)_{1 \le i,j \le n}$ and $\mathbf{D}^{(k)} = \text{diag}(\| \mathbf{P}_{1.}^{(k)} \|_2, \dots, \| \mathbf{P}_{n.}^{(k)} \|_2)$.

2. Obtain eigenscores: for each $i \in \{1, 2, \dots, n\}$,

   (i) calculate the similarity matrix

$$\mathbf{G}_i = ((\bar{\mathbf{P}}_{i.}^{(k_1)})^{\top} \bar{\mathbf{P}}_{i.}^{(k_2)})_{1 \le k_1, k_2 \le K}. \qquad (6)$$

   (ii) perform eigen-decomposition of $\mathbf{G}_i$ and define the eigenscores

$$\hat{\mathbf{s}}_i = (\hat{s}_{i,1}, \hat{s}_{i,2}, \dots, \hat{s}_{i,K}) := |\hat{\mathbf{u}}_i|, \qquad (7)$$

where $\hat{\mathbf{u}}_i$ is the eigenvector of $\mathbf{G}_i$ associated to its largest eigenvalue.

3. Construct meta-distance matrix: for each $i \in \{1, 2, \dots, n\}$, calculate the eigenscore-weighted average

$$\bar{\mathbf{P}}_{i.}^m = \sum_{k=1}^{K} \hat{s}_{i,k} \bar{\mathbf{P}}_{i.}^{(k)}, \qquad (8)$$

and define $\bar{\mathbf{P}}^m \in \mathbb{R}^{n \times n}$ whose $i$-th row is $\bar{\mathbf{P}}_{i.}^m$.

4. Obtain meta-visualization: apply an existing visualization method (e.g., UMAP or kPCA) to $\bar{\mathbf{P}}^m$ to obtain a meta-visualization.

Output: the eigenscores $\{\hat{\mathbf{s}}_i\}_{1 \le i \le n}$, and the meta-visualization.

## Measuring normalized distances from each visualization

In order that the proposed method is invariant to the respective scale and coordinate basis (i.e., directionality) of the low-dimensional embeddings generated from different visualization method, we start by considering the normalized pairwise-distance matrix for each visualization.

Specifically, for each $k \in \{1, 2, \ldots, K\}$, we define the normalized pairwise-distance matrix

$$\bar{\mathbf{P}}^{(k)} = [\mathbf{D}^{(k)}]^{-1} \mathbf{P}^{(k)} \in \mathbb{R}^{n \times n}, \tag{9}$$

where

$$\mathbf{P}^{(k)} = (\| \mathbf{X}_i^{(k)} - \mathbf{X}_j^{(k)} \|_2)_{1 \le i,j \le n} \in \mathbb{R}^{n \times n}, \tag{10}$$

is the un-normalized Euclidean distance matrix, and $\mathbf{D}^{(k)} = \mathrm{diag}\,(\| \mathbf{P}_{1.}^{(k)} \|_2, \| \mathbf{P}_{2.}^{(k)} \|_2, \ldots, \| \mathbf{P}_{n.}^{(k)} \|_2)$ is a diagonal matrix with its diagonal entries being the $\ell_2$-norms of the rows $\{\mathbf{P}_{1.}^{(k)}, \ldots, \mathbf{P}_{n.}^{(k)}\}$ of $\mathbf{P}^{(k)}$. As a result, the normalized distance matrix $\bar{\mathbf{P}}^{(k)}$ has its rows being unit vectors, and is invariant to any scaling and rotation of the visualization $\{\mathbf{X}_i^{(k)}\}_{1 \le i \le n}$.

The normalized distance matrices $\{\bar{\mathbf{P}}^{(k)}\}_{1 \le k \le K}$ summarize the candidate visualizations in a compact and efficient way. Their scale- and rotation-invariance properties are particularly useful for comparing visualizations produced by distinct methods.

## Sample-wise eigenscores for assessing visualizations

Our spectral method for assessing multiple visualizations is based on the normalized distance matrices $\{\bar{\mathbf{P}}^{(k)}\}_{1 \le k \le K}$. For each $i \in \{1, 2, \ldots, n\}$, we define the similarity matrix

$$\mathbf{G}_i = ((\bar{\mathbf{P}}_{i.}^{(k_1)})^\top \bar{\mathbf{P}}_{i.}^{(k_2)})_{1 \le k_1, k_2 \le K} \in \mathbb{R}^{K \times K}, \tag{11}$$

which summarizes the pairwise similarity between the candidate visualizations with respect to sample $i$. By construction, the entries of $\mathbf{G}_i$ are inner-products between unit vectors, each representing the normalized distances associated with sample $i$ in a candidate visualization. Naturally, a larger entry $(\bar{\mathbf{P}}_{i.}^{(k_1)})^\top \bar{\mathbf{P}}_{i.}^{(k_2)}$ indicates higher concordance between the two candidate visualizations. Then, for each $i \in \{1, 2, \ldots, n\}$, we define the vector of eigenscores $\hat{\mathbf{s}}_i = (\hat{s}_{i,1}, \ldots, \hat{s}_{i,K})$ for the candidate visualizations with respect to sample $i$ as the absolute value of the eigenvector $\hat{\mathbf{u}}_i \in \mathbb{R}^K$ of $\mathbf{G}_i$ associated to its largest eigenvalue, that is,

$$\hat{\mathbf{s}}_i : = |\hat{\mathbf{u}}_i|, \tag{12}$$

where the absolute value function $|\cdot|$ is applied entrywise. As explained in our main text, the nonnegative components of $\hat{\mathbf{s}}_i$ quantify the relative performance of $K$ candidate visualizations with respect to sample $i$, with higher eigenscores indicating better performance. Consequently, for each candidate visualization $\{\mathbf{X}_i^{(k)}\}_{1 \le i \le n}$, one obtains a set of eigenscores $\{\hat{s}_{i,k}\}_{1 \le i \le n}$ summarizing its performance relative to other candidate visualizations in a sample-wise manner. Ranking and selection among candidate visualizations can be achieved based on various summary statistics of the eigenscores, such as mean, median, or coefficient of variation, depending on the specific applications. In particular, when some candidate visualizations are produced by the same method but under different tuning parameters, the eigenscores can be used to select the most suitable tuning parameters for visualizing the dataset. However, a more substantial application of the eigenscores is to combine multiple data visualizations into a meta-visualization, which has improved signal-to-noise ratio and higher resolution of the structural information contained in the data.

Importantly, the eigenscores essentially take the underlying true signals rather than the noisy observations $\{\mathbf{Y}_i\}_{1 \le i \le n}$ as its referential target for performance assessment, making the method easier to implement and less susceptible to the effect of noise in the original data (Supplementary Figure 14).

## Meta-visualization using eigenscores

Using the above eigenscores, one can construct a meta-distance matrix properly combining the information contained in each candidate visualization. Specifically, for each $i \in \{1, 2, \ldots, n\}$, we define the vector of meta-distances with respect to sample $i$ as the eigenscore-weighted average of all the normalized distances respect to sample $i$, that is,

$$\bar{\mathbf{P}}_{i.}^m = \sum_{k=1}^K \hat{s}_{i,k} \bar{\mathbf{P}}_{i.}^{(k)} \in \mathbb{R}^n. \tag{13}$$

Then, the meta-distance matrix is defined as $\bar{\mathbf{P}}^m \in \mathbb{R}^{n \times n}$ whose $i$-th row is $\bar{\mathbf{P}}_{i.}^m$. To obtain a meta-visualization, we take the meta-distance matrix $\bar{\mathbf{P}}^m$ and apply an existing visualization method that allows for the meta-distance $\bar{\mathbf{P}}^m$ (or its symmetrized version $\bar{\mathbf{P}}^m + (\bar{\mathbf{P}}^m)^\top$) as its input.

Intuitively, for each $i = 1, 2, \ldots, n$, we essentially apply a principal component (PC) analysis to the normalized distance matrix $\bar{\mathbf{P}}_i = \begin{bmatrix} \bar{\mathbf{P}}_{i.}^{(1)} & \bar{\mathbf{P}}_{i.}^{(2)} & \ldots & \bar{\mathbf{P}}_{i.}^{(K)} \end{bmatrix} \in \mathbb{R}^{n \times K}$. Specifically, by definition[45] the leading eigenvector $\hat{\mathbf{u}}_i$ of $\mathbf{G}_i = \bar{\mathbf{P}}_i^\top \bar{\mathbf{P}}_i \in \mathbb{R}^{K \times K}$ is the first PC loadings of $\bar{\mathbf{P}}_i$, whereas the first PC is defined as the linear combination $\bar{\mathbf{P}}_i \hat{\mathbf{u}}_i = \sum_{k=1}^K \hat{u}_{i,k} \bar{\mathbf{P}}_{i.}^{(k)}$. Under the condition that the first PC loadings are all nonnegative (which is ensured with high probability under condition (C2) below), the first PC $\bar{\mathbf{P}}_i \hat{\mathbf{u}}_i$ is exactly the meta-distance $\bar{\mathbf{P}}_{i.}^m$ defined in (13) above. When interpreted as PC loadings, the leading eigenvector $\hat{\mathbf{u}}_i$ of $\mathbf{G}_i$ contains weights for different vectors $\{\bar{\mathbf{P}}_{i.}^{(k)}\}_{1 \le k \le K}$ so that the final linear combination $\bar{\mathbf{P}}_i \hat{\mathbf{u}}_i$ has the largest variance, that is, summarizes the most information contained in $\bar{\mathbf{P}}_i$. It is in this sense that the meta-distance $\bar{\mathbf{P}}_{i.}^m$ is a consensus across $\{\bar{\mathbf{P}}_{i.}^{(k)}\}_{1 \le k \le K}$.

For our own numerical studies, we used UMAP for meta-visualizing datasets with cluster structures, and used kPCA for meta-visualizing all the other datasets with smoother manifold structures, such as trajectory, cycle, or mixed structures. The choice of UMAP in the former case was due to its advantage in treating large numbers of clusters without requiring prior knowledge about the number of clusters[6,16]; whereas the choice of kPCA in the latter case was rooted in its advantage in capturing nonlinear smooth manifold structures[46]. In each case, the hyper-parameters used for generating the meta-visualization were determined without further tuning – for example, when using UMAP for meta-visualization, we set the hyper-parameters the same as those associated to the UMAP visualization which achieved higher median eigenscore than other UMAP visualizations. Moreover, while in general UMAP/kPCA works well as a default method for meta-visualization, our proposed algorithm is robust with respect to the choice of this final visualization method. In our numerical analysis, we observed empirically that other methods such as t-SNE and PHATE could also lead to meta-visualizations with comparably substantial improvement over individual candidate visualizations in terms of the concordance with the underlying true low-dimensional structure of the data (see Supplementary Figure 6). In addition, the meta-visualization shows robustness to potential outliers in the data (Figs. 4 and 5).

Under a generic signal-plus-noise model, we obtain explicit theoretical conditions under which the performance of the proposed spectral method is guaranteed. These conditions provide proper interpretations and guidance on the application of the method, such as how to more effectively prepare the candidate visualizations. For

better understanding, we summarize these technical conditions informally as follows.

(C1′) The performance of candidate visualizations are sufficiently diverse in terms of their individual distortions from the underlying true structures.

(C2′) The candidate visualizations altogether contains sufficient amount of information about the underlying true structures.

Intuitively, Condition (C1′) concerns diversity of methods in producing candidate visualizations, whereas Condition (C2′) is related to the quality of the candidate visualizations. In practice, Condition (C2′) is satisfied when the signal-to-noise ratio in the data, as described by (2), is sufficiently large, so that the adopted visualization methods perform reasonably well on average. On the other hand, a sufficient condition for (C1′) is that, at most $\sqrt{K}$ out of $K$ candidate visualizations are very similarly distorted from the true patterns in terms of the normalized distances $\bar{\mathbf{P}}^{(k)}$. This would allow, for example, groups of up to 3 to 4 candidate visualizations out of 10 to 15 visualizations being produced by very similar procedures such as the same method under different hyper-parameters.

### Simulations
For each simulation setting, we let the diameter of the underlying structure vary within a certain range so that the final results are comparable across different structures. The final boxplots in Fig. 2a, and Supplementary Figures 2 and 4 summarize the simulation results across 20 equispaced diameter values for each underlying structure.

### Religious and biblical texts data
Each visualization method was applied to the Document Term Matrix, with 8265 centred and normalized features and 590 samples (text fragments). The raw data are provided in the Source Data file. For the 16 PHATE-based candidate visualizations obtained in Supplementary Figure 15, we consider 16 values of nearest neighbor parameter knn ranging from 2 to 150 with equal space.

### Cell cycle data
The raw count data were preprocessed, normalized, and scaled by following the standard procedure (R functions CreateSeuratObject, NormalizeData and ScaleData under default settings) as incorporated in the R package Seurat. We also applied the R function FindVariableFeatures in Seurat to identify 2000 most variable genes for subsequent analysis. The final $p = 1147$ cell-cycle related genes were selected based on two-sample t-tests. The preprocessed data are provided in the Source Data file. The 16 candidate visualizations were generated the same way as in the previous example.

### Cell differentiation data
The raw count data were preprocessed, normalized, and scaled using `Seurat` package by following the similar procedure as described above. The pre-processed data are provided in the Source Data file. The 16 candidate visualizations were generated the same way as in the previous examples.

### Computational cost
We considered the single-cell transcriptomic dataset[43] that contains more than 20,000 cells of different cell types from the neurogenic regions of 28 mice. For each $n \in \{1000, 2000, 4000, 8000, 14000\}$, we randomly select $n$ cells of nine different cell types, and selected subsets of $p \in \{500, 1000, 2000\}$ genes to obtain an $n \times p$ count matrix. After normalizing the count matrix, we applied various visualization methods (PCA, HLLE, kPCA, LEIM, UMAP, t-SNE and PHATE) that are in general scalable to large datasets (i.e., cost less than one minute for

visualizing 1000 samples of dimension 300), to generate 11 candidate visualizations (with two different parameter settings for kPCA, t-SNE, UMAP and PHATE). Then we ran our proposed algorithm to obtain the final meta-visualization.

### Reporting summary
Further information on research design is available in the Nature Portfolio Reporting Summary linked to this article.

## Data availability
The religious and biblical text data[38] are downloaded from UCI Machine Learning Repository [https://archive.ics.uci.edu/ml/machine-learning-databases/00512/]. The cell cycle analysis is based on the mouse embryonic stem cell data[40] available in EMBL-EBI with accession code [E-MTAB-2805]. The cell trajectory analysis is based on the mouse embryonic stem single cell data[42], available in Gene Expression Omnibus with accession code [GSE98664]. The single-cell transcriptomic dataset[43] used for evaluating computational cost is accessible at BioProject with accession code [PRJNA795276]. Source data are provided with this paper.

## Code availability
The R codes of the method, and for reproducing our simulations and data analyses are available at our GitHub repository meta-visualization[47] https://github.com/rongstat/meta-visualization.

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

## Acknowledgements

R.M. would like to thank David Donoho and Rui Duan for helpful discussions. Funding support was provided by Professor David Donoho at Stanford University (R.M.), Knight-Hennessy Scholars program (E.D.S.), Paul and Daisy Soros Fellowship for New Americans (E.D.S.), the National Science Foundation Graduate Research Fellowship Program (E.D.S.), NSF CAREER 1942926 (J.Z.), NIH P30AG059307 (J.Z.), 5RM1HG010023 (J.Z.) and grants from the Silicon Valley Foundation (J.Z.) and the Chan-Zuckerberg Initiative (J.Z.).

## Author contributions

R.M. and J.Z. conceived of the study. R.M. designed and implemented the method with input from J.Z. and E.D.S. E.D.S. contributed to the numerical analysis and software implementation. R.M. designed and developed the theoretical results for the study. R.M. prepared a draft of the manuscript. J.Z. and E.D.S. edited the manuscript.

## Competing interests

The Authors declare no competing interests.
