## [Peer Review File · Nature Communications]

A Spectral Method for Assessing and Combining Multiple Data VisualizationsREVIEWER COMMENTS

Reviewer #1 (Remarks to the Author):

The paper proposes a new method of combining disparate techniques for visualization of low dimensional structures hidden in high-dimensional images into a single representative visualization tool. The representation is built upon the notion of "eigenscore", namely the unsigned leading eigenvector of the inner product matrix of normalized pairwise distances among features corresponding to each sample observation. These eigenscores are used to reweight the normalized pairwise distances from each observation to the rest to obtain the a meta-distance matrix. Finally, commonly used visualization tools such as kernel PCA or uniform manifold approximation is applied to the meta-distance matrix to obtain the final meta-visualization.

The authors make a compelling argument, by illustrating the proposed technique through numerous real-life and simulated data, that the proposed meta-visualization technique is more representative of the underlying structure than what is captured by individual constituent visualization tools.

They also provide some theoretical support, in terms of noise removal characteristics, of the proposed eigenscore-based representation scheme when the data are noisy.

There are two pertinent questions that the authors should address carefully.

1. Why does the leading eigenvector of G_i yield the most informative representation of the different visualization schemes associated with the i -th sample? What about the eigenvectors associated with the non-leading eigenvalues?

2. The behavior of $\{h_i^{(k)}\}$ should be determined by the noise vectors $\{Z_i\}$. It is not clear why a separate assumption (C1a) on the former is needed, or indeed what exactly is the scope of its validity. To clarify, it would be good to know under what assumptions on the noise vectors $\{Z_i\}$, assumption (C1a), coupled with the representation (4.2) is valid.

Reviewer #2 (Remarks to the Author):

The work proposes a novel methodology to combine dimensionality reduction embeddings to produce more reliable and robust high-dimensional data visualizations. The work innovates in three main aspects: 1) the methodology relies only on the embeddings produced by a set of dimensionality reduction methods, disregarding the original data altogether. Although previous techniques such as Projection Inspector [1] (authors should include this reference in Sec. 1.1) can also combine embeddings without considering the original data, the mechanism proposed in the present paper provides a consensus visualization that reflects the local concordance of the underlying low-dimensional structures, a characteristic not present in previous works. 2) The proposed eigenscore-based weighting scheme allows for locally evaluating the performance of a dimensionality reduction method relative to other embeddings, which is quite interesting and valuable. 3) The theoretical asymptotic analysis brings robustness to the proposed method, guaranteeing the performance of the "meta-visualization".

Despite the noteworthy results, the work bears a number of flaws that must be addressed before being considered for publication. For instance, there is no discussion of why the eigenscores do the trick. The eigenvector associated with the largest eigenvalue of G_i can be seen as the principal component of normalized distances; therefore, the eigenscores are related to loadings. If I am right, this is why the eigenscores-based weights properly combine the embeddings. The authors should include some intuitive discussion to better convince about the effectiveness of the eigenscores when

averaging the embeddings.

Regarding the simulation studies (Sec 3.1), as the intrinsic dimension of the datasets are $r=6,2,3$, it doesn't make sense to set $p=500$ and $p=300$, as the remaining dimensions are filled with zeros. Therefore, the claim that the experimental setup involves high-dimensional data is not true and must be rephrased.

The color map in Fig. 2a) makes it pretty hard to figure out which curve corresponds to each technique. I understand that finding proper colors is not easy, but as it is, Fig. 2a) is not readable.

The computational cost is an important issue with the proposed method. In sec. 3.5, authors claim the whole pipeline took at most two seconds to run in all experiments, faster than computing the embeddings. The reason for that is the small size of the datasets. Eigenvector computation is costly, mainly when dealing with full matrices (no sparsity), as is the case here. Therefore, the discussion in sec. 3.5 is useless. As the authors acknowledge, the proposed methodology does not scale, so it is essential to provide information about how far one can go regarding dataset size and data dimension. Authors should generate multiple datasets with varying sizes and dimensions, measuring the computational performance in different scenarios.

In summary, the work brings novel and interesting ideas that are theoretically supported. However, the applicability of the proposed methodology in problems involving mid to large high-dimensional datasets is questionable (if not impracticable). In other words, the scope of applications is limited.

[1] Pagliosa, Paulo, et al. "Projection Inspector: Assessment and synthesis of multidimensional projections." *Neurocomputing* 150 (2015): 599-610.

Point-by-Point Response to the Referee 1

NCOMMS-22-30102

We would like to thank you for your careful reading of the paper and the helpful comments and suggestions. Please see below our point-by-point response to your remarks and the changes we made.

Comment 1.1: *The paper proposes a new method of combining disparate techniques for visualization of low dimensional structures hidden in high-dimensional images into a single representative visualization tool. The representation is built upon the notion of “eigen-score”, namely the unsigned leading eigenvector of the inner product matrix of normalized pairwise distances among features corresponding to each sample observation. These eigen-scores are used to reweight the normalized pairwise distances from each observation to the rest to obtain the a meta-distance matrix. Finally, commonly used visualization tools such as kernel PCA or uniform manifold approximation is applied to the meta-distance matrix to obtain the final meta-visualization. The authors make a compelling argument, by illustrating the proposed technique through numerous real-life and simulated data, that the proposed meta-visualization technique is more representative of the underlying structure than what is captured by individual constituent visualization tools. They also provide some theoretical support, in terms of noise removal characteristics, of the proposed eigenscore-based representation scheme when the data are noisy.*

Thank you so much for your appreciation of our paper!

Comment 1.2: *Why does the leading eigenvector of \mathbf{G}_i yield the most informative representation of the different visualization schemes associated with the i -th sample? What about the eigenvectors associated with the non-leading eigenvalues?*

Thanks for the great question. Intuitively, for each $i = 1, 2, \dots, n$, one could consider our approach as doing a principal component analysis on the normalized distance matrix

$$\bar{\mathbf{P}}_i = \begin{bmatrix} \bar{\mathbf{P}}_i^{(1)} & \bar{\mathbf{P}}_i^{(2)} & \dots & \bar{\mathbf{P}}_i^{(K)} \end{bmatrix} \in \mathbb{R}^{n \times K}, \quad (0.1)$$

over K candidate visualizations. Specifically, by definition (Jolliffe & Cadima, 2016), the leading eigenvector $\hat{\mathbf{u}}_i$ of the product matrix $\mathbf{G}_i = \bar{\mathbf{P}}_i^\top \bar{\mathbf{P}}_i \in \mathbb{R}^{K \times K}$ is the first PC loadings of $\bar{\mathbf{P}}_i$, whereas the first principal component (PC1) is defined as the linear combination $\bar{\mathbf{P}}_i \hat{\mathbf{u}}_i = \sum_{k=1}^K \hat{u}_{i,k} \bar{\mathbf{P}}_i^{(k)}$, with its elements commonly referred as the PC scores associated to PC1, as they are the values that each sample would score along PC1. Under the condition that the first PC loadings are all nonnegative (which is ensured with high probability under

condition (C2)), the first principal component $\bar{\mathbf{P}}_i \hat{\mathbf{u}}_i$, is exactly the meta-distance $\bar{\mathbf{P}}_i^m$ defined in (2.9) of the revised manuscript. Note that

$$\hat{\mathbf{u}}_i = \arg \max_{\mathbf{u} \in \mathbb{R}^K, \|\mathbf{u}\|_2=1} \mathbf{u}^\top \mathbf{G}_i \mathbf{u} = \arg \max_{\mathbf{u} \in \mathbb{R}^K, \|\mathbf{u}\|_2=1} \|\bar{\mathbf{P}}_i \mathbf{u}\|_2^2. \quad (0.2)$$

When interpreted as PC loadings, the leading eigenvector $\hat{\mathbf{u}}_i$ of \mathbf{G}_i contains weights for different vectors $\{\bar{\mathbf{P}}_i^{(k)}\}_{1 \leq k \leq K}$ so that the final linear combination $\bar{\mathbf{P}}_i \hat{\mathbf{u}}_i$, according to (0.2), has the largest variance, that is, summarizes the most information contained in $\bar{\mathbf{P}}_i$. It is in this sense that the meta-distance $\bar{\mathbf{P}}_i^m$ is a consensus across $\{\bar{\mathbf{P}}_i^{(k)}\}_{1 \leq k \leq K}$. Similarly, for the eigenvectors associated with other non-leading eigenvalues of \mathbf{G}_i , they can be interpreted as the loadings of PC2, PC3, etc. However, by definition {PC2, PC3, ...} only contain the remaining information that is orthogonal to PC1. Thus, compared with PC1, they are less suitable to be treated as a consensus across $\{\bar{\mathbf{P}}_i^{(k)}\}_{1 \leq k \leq K}$.

The above clarifications have been added to **Section 2.1.3** of the revised manuscript.

Comment 1.3: *The behavior of $\mathbf{h}_i^{(k)}$ should be determined by the noise vectors \mathbf{Z}_i . It is not clear why a separate assumption (C1a) on the former is needed, or indeed what exactly is the scope of its validity. To clarify, it would be good to know under what assumptions on the noise vectors \mathbf{Z}_i , assumption (C1a), coupled with the representation (4.2) is valid.*

Thanks for the helpful comments and questions. In the following, we provide a sufficient condition with some intuitions that implies the sub-Gaussian condition (C1a) for $\mathbf{h}_i^{(k)}$, coupled with (2.12) of the revised manuscript (i.e., (4.2) of the previous version). In general, the distortion vector $\mathbf{h}_i^{(k)}$ is jointly determined by the noise $\{\mathbf{Z}_i\}_{1 \leq i \leq n}$, the noiseless samples $\{\mathbf{Y}_i^*\}_{1 \leq i \leq n}$ and the dimension reduction map $f_k : \mathbb{R}^p \rightarrow \mathbb{R}^2$ associated to the k -th visualization method. Accordingly, our sufficient condition for (C1a) essentially involves regularity of the signal structure $\{\mathbf{Y}_i^*\}_{1 \leq i \leq n}$ and the dimension reduction (DR) maps $f_k : \mathbb{R}^p \rightarrow \mathbb{R}^2$ for $1 \leq k \leq K$, and sub-Gaussianity of the noise vector $\{\mathbf{Z}_i\}_{1 \leq i \leq n}$. We first state precisely our sufficient condition.

(C01) (*Regularity of the signal and DR map*) The noiseless samples $\{\mathbf{Y}_i^*\}_{1 \leq i \leq n}$ lie on a bounded manifold \mathcal{M} embedded in \mathbb{R}^p , and the DR map f_k and its first-order derivative are bounded in the sense that for all $\mathbf{Y} \in \mathbb{R}^p$

$$L^{-1} \leq \|f_k(\mathbf{Y})\|_2 \leq L, \quad \left\| \frac{\partial f_k(\mathbf{Y})}{\partial \mathbf{Y}} \right\| \leq C_f, \quad (0.3)$$

almost surely for some constants $L > 1$ and $C_f > 0$.

(C02) (*Sub-Gaussian noise*) The noise vectors $\{\mathbf{Z}_i\}_{1 \leq i \leq n}$ are sub-Gaussian random vectors.

Intuitively, (C01) requires that the underlying signal structure is finite and that the DR map is also finite and sufficiently smooth. In particular, we allow that f_k is random in itself, as in the cases of randomized algorithms such as t-SNE and UMAP. The sub-Gaussian condition (C02) on the noise vector \mathbf{Z}_i is mild and allows for wide range of noise structures.

Below we show that Conditions (C01) and (C02) jointly imply the sub-Gaussianity of $\mathbf{h}_i^{(k)}$. Firstly, note that by definition $\mathbf{P}_i^{(k)} = (\|f_k(\mathbf{Y}_i) - f_k(\mathbf{Y}_1)\|_2, \|f_k(\mathbf{Y}_i) - f_k(\mathbf{Y}_2)\|_2, \dots, \|f_k(\mathbf{Y}_i) - f_k(\mathbf{Y}_n)\|_2)^\top$. Then for each i , we can define the pairwise distance for the noiseless samples associated with the k -th visualization method as

$$\mathbf{P}_i^{*(k)} = (\|f_k(\mathbf{Y}_i^*) - f_k(\mathbf{Y}_1^*)\|_2, \|f_k(\mathbf{Y}_i^*) - f_k(\mathbf{Y}_2^*)\|_2, \dots, \|f_k(\mathbf{Y}_i^*) - f_k(\mathbf{Y}_n^*)\|_2)^\top. \quad (0.4)$$

Recall that $\mathbf{P}_i^* = (\|\mathbf{Y}_i^* - \mathbf{Y}_1^*\|_2, \|\mathbf{Y}_i^* - \mathbf{Y}_2^*\|_2, \dots, \|\mathbf{Y}_i^* - \mathbf{Y}_n^*\|_2)^\top$. Then, by (4.2) of the revised manuscript, it follows that

$$\begin{aligned} \mathbf{h}_i^{(k)} &= c_k^{-1} \mathbf{P}_i^{(k)} - \mathbf{P}_i^* \\ &= (c_k^{-1} \mathbf{P}_{ij}^{*(k)} + \frac{c_k^{-1} \mathbf{g}_{k,ij}^\top [f_k(\mathbf{Y}_i) - f_k(\mathbf{Y}_j) - f_k(\mathbf{Y}_i^*) - f_k(\mathbf{Y}_j^*)]}{\|\mathbf{g}_{k,ij}\|_2} - \mathbf{P}_{ij}^*)_{1 \leq j \leq n} \end{aligned} \quad (0.5)$$

$$= \left(c_{i,k}^{-1} \mathbf{P}_{ij}^{*(k)} - \mathbf{P}_{ij}^* + \frac{c_{i,k}^{-1} \mathbf{g}_{k,ij}^\top \left[\frac{\partial f_k(\mathbf{Y})}{\partial \mathbf{Y}} \Big|_{\mathbf{Y}=\mathbf{s}_i} \mathbf{Z}_i - \frac{\partial f_k(\mathbf{Y})}{\partial \mathbf{Y}} \Big|_{\mathbf{Y}=\mathbf{s}_j} \mathbf{Z}_j \right]}{\|\mathbf{g}_{k,ij}\|_2} \right)_{1 \leq j \leq n}, \quad (0.6)$$

where in (0.5) we used Taylor expansion of $\mathbf{P}_{ij}^{(k)} = \|f_k(\mathbf{Y}_i) - f_k(\mathbf{Y}_j)\|_2$ at $f_k(\mathbf{Y}_i^*) - f_k(\mathbf{Y}_j^*)$ with $\mathbf{g}_{k,ij}$ being some point between $\mathbf{P}_{ij}^{(k)}$ and $\mathbf{P}_{ij}^{*(k)}$, and in (0.6) we used Taylor expansion of $f_k(\mathbf{Y}_i)$ at \mathbf{Y}_i^* with \mathbf{s}_i being some point between \mathbf{Y}_i and \mathbf{Y}_i^* .

For each i , since $c_{i,k}$ is a parameter that accounts for the possible scaling difference caused by the dimension reduction map f_k , without loss of generality, we can take $c_{i,k} = \frac{\|f_k(\mathbf{Y}_i^*)\|_2}{\|\mathbf{Y}_i^*\|_2}$. Under Condition (C01), it follows that $c_k^{-1} \mathbf{P}_{ij}^{*(k)} - \mathbf{P}_{ij}^*$ in (0.6) is bounded and therefore a sub-Gaussian random variable. Similarly, for the random variable

$$\frac{c_{i,k}^{-1} \mathbf{g}_{k,ij}^\top \left[\frac{\partial f_k(\mathbf{Y})}{\partial \mathbf{Y}} \Big|_{\mathbf{Y}=\mathbf{s}_i} \mathbf{Z}_i - \frac{\partial f_k(\mathbf{Y})}{\partial \mathbf{Y}} \Big|_{\mathbf{Y}=\mathbf{s}_j} \mathbf{Z}_j \right]}{\|\mathbf{g}_{k,ij}\|_2} \quad (0.7)$$

in (0.6), under Condition (C01), we also have the boundedness of $\left\| \frac{\partial f_k(\mathbf{Y})}{\partial \mathbf{Y}} \Big|_{\mathbf{Y}=\mathbf{s}_i} \right\|$ and $\left\| \frac{\partial f_k(\mathbf{Y})}{\partial \mathbf{Y}} \Big|_{\mathbf{Y}=\mathbf{s}_j} \right\|$. By Proposition 2.5.2 of Vershynin (2018), these along with the boundedness of $c_{i,k}^{-1}$ and $\frac{\mathbf{g}_{k,ij}}{\|\mathbf{g}_{k,ij}\|_2}$, and the sub-Gaussianity of \mathbf{Z}_i and \mathbf{Z}_j from (C02), imply that (0.7) is also a sub-Gaussian random variable. Thus, we have verified that the sub-Gaussianity of $\mathbf{h}_i^{(k)}$ under Conditions (C01) and (C02). In particular, the sub-Gaussian parameter σ^2 in (C1a) is jointly determined by the underlying manifold \mathcal{M} , the scaling (L) and the smoothness (C_f) of the DR map.

These clarifications have been added to **Section 2.3.1** of the revised manuscript, as well as in **Section B.2** of the revised Supplement.

Point-by-Point Response to the Referee 2

We would like to thank you for careful reading of the paper and your helpful comments and suggestions. Please see below our point-by-point responses to your remarks and the changes we made.

Comment 2.1: *The work proposes a novel methodology to combine dimensionality reduction embeddings to produce more reliable and robust high-dimensional data visualizations. The work innovates in three main aspects: 1) the methodology relies only on the embeddings produced by a set of dimensionality reduction methods, disregarding the original data altogether. Although previous techniques such as Projection Inspector [1] (authors should include this reference in Sec. 1.1) can also combine embeddings without considering the original data, the mechanism proposed in the present paper provides a consensus visualization that reflects the local concordance of the underlying low-dimensional structures, a characteristic not present in previous works. 2) The proposed eigenscore-based weighting scheme allows for locally evaluating the performance of a dimensionality reduction method relative to other embeddings, which is quite interesting and valuable. 3) The theoretical asymptotic analysis brings robustness to the proposed method, guaranteeing the performance of the "meta-visualization".*

Thank you very much for appreciation of our paper, and thanks for pointing out the interesting paper by Pagliosa et al (2015). In our revised manuscript, we have included this reference in **Section 1.1**.

Comment 2.2: *Despite the noteworthy results, the work bears a number of flaws that must be addressed before being considered for publication. For instance, there is no discussion of why the eigenscores do the trick. The eigenvector associated with the largest eigenvalue of G_i can be seen as the principal component of normalized distances; therefore, the eigenscores are related to loadings. If I am right, this is why the eigenscores-based weights properly combine the embeddings. The authors should include some intuitive discussion to better convince about the effectiveness of the eigenscores when averaging the embeddings.*

Thanks for the comment and suggestion! You are correct. Intuitively, for each $i = 1, 2, \dots, n$, one could consider our approach as doing a principal component analysis on the normalized distance matrix

$$\bar{\mathbf{P}}_i = \begin{bmatrix} \bar{\mathbf{P}}_i^{(1)} & \bar{\mathbf{P}}_i^{(2)} & \dots & \bar{\mathbf{P}}_i^{(K)} \end{bmatrix} \in \mathbb{R}^{n \times K}, \quad (0.8)$$

over K candidate visualizations. Specifically, by definition (Jolliffe & Cadima, 2016), the leading eigenvector $\hat{\mathbf{u}}_i$ of the product matrix $\mathbf{G}_i = \bar{\mathbf{P}}_i^\top \bar{\mathbf{P}}_i \in \mathbb{R}^{K \times K}$ is the first PC loadings of $\bar{\mathbf{P}}_i$, whereas the first principal component (PC1) is defined as the linear combination

$\bar{\mathbf{P}}_i \hat{\mathbf{u}}_i = \sum_{k=1}^K \hat{u}_{i,k} \bar{\mathbf{P}}_i^{(k)}$, with its elements commonly referred as the PC scores associated to PC1, as they are the values that each sample would score along PC1. Under the condition that the first PC loadings are all nonnegative (which is ensured with high probability under condition (C2)), the first principal component $\bar{\mathbf{P}}_i \hat{\mathbf{u}}_i$, is exactly the meta-distance $\bar{\mathbf{P}}_i^m$ defined in (2.9) of the revised manuscript. Note that

$$\hat{\mathbf{u}}_i = \arg \max_{\mathbf{u} \in \mathbb{R}^K, \|\mathbf{u}\|_2=1} \mathbf{u}^\top \mathbf{G}_i \mathbf{u} = \arg \max_{\mathbf{u} \in \mathbb{R}^K, \|\mathbf{u}\|_2=1} \|\bar{\mathbf{P}}_i \mathbf{u}\|_2^2. \quad (0.9)$$

When interpreted as PC loadings, the leading eigenvector $\hat{\mathbf{u}}_i$ of \mathbf{G}_i contains weights for different vectors $\{\bar{\mathbf{P}}_i^{(k)}\}_{1 \leq k \leq K}$ so that the final linear combination $\bar{\mathbf{P}}_i \hat{\mathbf{u}}_i$, according to (0.9) has the largest variance, that is, summarizes the most information contained in $\bar{\mathbf{P}}_i$. It is in this sense that the meta-distance $\bar{\mathbf{P}}_i^m$ is a consensus across $\{\bar{\mathbf{P}}_i^{(k)}\}_{1 \leq k \leq K}$. Similarly, for the eigenvectors associated with other non-leading eigenvalues of \mathbf{G}_i , they can be interpreted as the loadings of PC2, PC3, etc. However, by definition {PC2, PC3, ...} only contain the remaining information that is orthogonal to PC1. Compared with PC1, they are apparently less suitable to be treated as a consensus across $\{\bar{\mathbf{P}}_i^{(k)}\}_{1 \leq k \leq K}$. This is why we used PC1 and its loadings for constructing the meta-distance.

The above clarifications have been added to **Section 2.1.3** of the revised manuscript.

Comment 2.3: *Regarding the simulation studies (Sec 3.1), as the intrinsic dimension of the datasets are $r=6,2,3$, it doesn't make sense to set $p=500$ and $p=300$, as the remaining dimensions are filled with zeros. Therefore, the claim that the experimental setup involves high-dimensional data is not true and must be rephrased.*

Thanks for the comments! We are sorry for the confusion and would like to make the following clarifications. Firstly, in the simulations of our previous manuscript, setting the remaining $(p - r)$ coordinates as zeros was just for convenience and was not due to any limitations of our algorithm. In the updated version, we generated the signals (noiseless samples \mathbf{Y}_i^*) from various low-dimensional structures isometrically embedded in the p -dimensional Euclidean space. In particular, each of the low-dimensional structures lie in some r -dimensional linear subspace, but is subject to an arbitrary rotation in \mathbb{R}^p , so that the signals are no longer sparse vectors (that is, all the p coordinates of \mathbf{Y}_i^* are in general nonzero). In our simulations, for some given signal-to-noise ratio (SNR) parameter $\theta > 0$, we generate $\{\mathbf{Y}_i^*\}_{1 \leq i \leq n}$ uniformly from each of the following three structures:

- (i) Finite point mixture with $r = 5$: $\{\mathbf{Y}_i^*\}_{1 \leq i \leq n}$ are independently sampled from the discrete set $\{\gamma_1, \gamma_2, \dots, \gamma_{r+1}\} \subset \mathbb{R}^p$ with equal probability, where γ_i 's are arbitrary orthogonal vectors in \mathbb{R}^p with the same length, i.e., $\|\gamma_i\|_2 = \theta$ for $1 \leq i \leq r + 1$.
- (ii) ‘‘Smiley face’’ with $r = 2$: $\{\mathbf{Y}_i^*\}_{1 \leq i \leq n}$ are generated independently and uniformly from

a two-dimensional “smiley face” structure (Supplement Figure S1 left) of diameter θ , isometrically embedded in \mathbb{R}^p and subject to an arbitrary rotation.

- (iii) “Mammoth” manifold with $r = 3$: $\{\mathbf{Y}_i^*\}_{1 \leq i \leq n}$ are generated independently uniformly from a three-dimensional “mammoth” manifold (Supplement Figure S1 right) of diameter θ , isometrically embedded in \mathbb{R}^p and subject to an arbitrary rotation.

After that, we added p -dimensional Gaussian noise \mathbf{Z}_i to the signals and used the p -dimensional noisy vectors $\mathbf{Y}_i = \mathbf{Y}_i^* + \mathbf{Z}_i$ as the observed data. The thus generated datasets cover diverse structures including Gaussian mixture clusters (i), mixed-type nonlinear clusters (ii), and a connected smooth manifold (iii).

Secondly, most dimension reduction or data visualization algorithms assume that the data contains some intrinsically low-dimensional structure embedded in the high-dimensional observation space, and aim at capturing such a low-dimensional structure. This is reflected in our simulation setup where r is relatively small and p is relatively large. In particular, after adding the noise \mathbf{Z}_i , the final observations $\{\mathbf{Y}_i\}_{1 \leq i \leq n}$ no longer lie in any low-dimensional subspace and become intrinsically p -dimensional. Our simulation shows that the meta-visualization better captures the underlying low-dimensional structure associated with \mathbf{Y}_i^* , compared to the candidate visualizations. Moreover, to demonstrate the flexibility of our method with respect to lower-to-higher values of intrinsic dimension r , in our revised version, under the cluster setting (i) above, we further evaluated the performance of different methods under various values of r , ranging from 5 to 50.

These clarifications have been incorporated in **Section 2.2.1** of the revised manuscript and **Section A.2** of the revised Supplement. In particular, the numerical results concerning the updated dense-signal simulation setup, described above, were summarized in **Figure 2** of the revised manuscript and **Figures S2 and S3** of the revised Supplement. The results for comparing the performance under different r values were summarized in **Figure S4** of the revised Supplement.

Comment 2.4: *The color map in Fig. 2a) makes it pretty hard to figure out which curve corresponds to each technique. I understand that finding proper colors is not easy, but as it is, Fig. 2a) is not readable.*

Thanks for pointing this out! In the revised version, to enable visual comparison across different methods, for each individual method, we replaced the lined-scatter plots in the previous version by a boxplot summarizing the mean concordance with the underlying true pattern across various signal-to-noise ratio parameter θ . The updated Figure 2(a) can be found on **page 11** of the revised manuscript. Similarly, we have updated **Figure S2** in the revised Supplement.

Comment 2.5: *The computational cost is an important issue with the proposed method. In sec. 3.5, authors claim the whole pipeline took at most two seconds to run in all experiments, faster than computing the embeddings. The reason for that is the small size of the datasets. Eigenvector computation is costly, mainly when dealing with full matrices (no sparsity), as is the case here. Therefore, the discussion in sec. 3.5 is useless. As the authors acknowledge, the proposed methodology does not scale, so it is essential to provide information about how far one can go regarding dataset size and data dimension. Authors should generate multiple datasets with varying sizes and dimensions, measuring the computational performance in different scenarios.*

Thanks for the comments and great suggestion! In terms of the computational cost, we would like to make the following clarifications. First, although our method relies on computing the leading eigenvector of generally non-sparse matrices, these matrices (i.e., \mathbf{G}_i in Algorithm 1 of the main paper) are of dimension $K \times K$, where K is the number of candidate visualizations, which is usually much smaller compared to the sample size n or dimensionality p of the original data. Thus, for each sample i , the computational cost due to the eigendecomposition is mild. Second, given the K candidate visualizations, our proposed algorithm is independent of the dimensionality p of the original dataset, as it only requires as input a set of $n \times 2$ low-dimensional embeddings produced by different visualization methods. Third, since our algorithm computes the eigenscores and the meta-distance with respect to each sample separately, the algorithm can be easily parallelized and carried out in multiple cores to further reduce time cost. We believe these features make the proposed algorithm readily scalable to relatively large and high-dimensional datasets.

Following your suggestion, we have included additional numerical evaluations of computational time for datasets with various sample sizes and dimensions. Specifically, we considered real single-cell transcriptomic datasets containing different cell types from the neurogenic regions of 28 mice (Buckley et al., 2022). For each $n \in \{1000, 2000, 4000, 8000, 14000\}$, we randomly select n cells of nine different cell types, and selected subsets of $p \in \{500, 1000, 2000\}$ genes to obtain an $n \times p$ count matrix. After normalizing the count matrix, we applied various visualization methods (PCA, HLLC, kPCA, LEIM, UMAP, t-SNE and PHATE) that are in general scalable to large datasets (i.e., costed less than one minute for visualizing 1000 samples of dimension 300), to generate 11 candidate visualizations (with two different parameter settings for kPCA, t-SNE, UMAP and PHATE). Then we ran our proposed algorithm to obtain the final meta-visualization. **Figure S15(b)** in the revised Supplement contains boxplots of median Silhouette indices for each candidate visualizations and the meta-visualization with respect to the underlying true cell types, showing the stable and superior performance of the proposed method under various sample sizes and dimensions. In **Figure S15(a)** of the revised Supplement, we compared the running time for generat-

ing the 11 candidate visualizations, and that for generating the meta-visualizations based on Algorithm 1, on a MacBook Pro with 2.2 GHz 6-Core Intel Core i7. In general, as n became large, the running time of the proposed algorithm also increased, but remained much less than that for generating the candidate visualizations. The difference in time cost became more significant as n increased, demonstrating that for large and high-dimensional datasets the computational cost essentially comes from generating candidate visualizations, rather than from the meta-visualization step. In particular, for dataset of sample size as large as 8000 and of dimension 2000, it took about 60 mins to generate all the 11 candidate visualizations, and took about additional 12 mins to generate the meta-visualization. Moreover, **Figure S15(a)** in the revised Supplement also confirmed that, for each n , when p increased, the running time for generating the candidate visualizations was longer, but the time cost for meta-visualization remained about the same.

The above clarifications and additional numerical results have been included in **Section 2.2.5** of the updated manuscript, and **Section A.2** of the revised Supplement.

Comment 2.6: *In summary, the work brings novel and interesting ideas that are theoretically supported. However, the applicability of the proposed methodology in problems involving mid to large high-dimensional datasets is questionable (if not impracticable). In other words, the scope of applications is limited.*

Thank you again for your very helpful feedback! We really appreciate your time. We hope that our response to your Comment 2.5 has clarified the scope of applicability of the proposed method, especially to large and high-dimensional datasets. Dimensional reduction and visualization are also often used on medium-size datasets of several thousand points (e.g. many human or clinical studies have less than a few hundred samples and many single-cell studies have a few thousand cells). Meta-visualization can be especially useful in these settings as there can be substantial variance across different visualizations of such data. Practitioners often create multiple visualizations for data exploration, and our approach can simply reuse these visualizations with little additional computational cost.

References

- Jolliffe, I. T., & Cadima, J. (2016). Principal component analysis: a review and recent developments. *Philosophical Transactions of the Royal Society A: Mathematical, Physical and Engineering Sciences*, 374(2065), 20150202.
- Pagliosa, P., Paulovich, F. V., Minghim, R., Levkowitz, H., & Nonato, L. G. (2015). Pro-

jection inspector: Assessment and synthesis of multidimensional projections. *Neurocomputing*, 150, 599-610.

Vershynin, R. (2018). *High-dimensional probability: An introduction with applications in data science*. Cambridge university press.

REVIEWERS' COMMENTS

Reviewer #2 (Remarks to the Author):

The authors have properly addressed all my comments and suggestions. I don't have any further comments. In my opinion, the manuscript is ready for publication.